# Temporal Slowness in Central Vision Drives Semantic Object Learning

**Timothy Schaumlöffel**[1,2,*], **Arthur Aubret**[3,4,*], **Gemma Roig**[1,2,†], **Jochen Triesch**[1,2,3,†]

[1]Department of Computer Science, Goethe University Frankfurt, Frankfurt am Main, Germany
[2]The Hessian Center for Artificial Intelligence (hessian.AI), Darmstadt, Germany
[3]Frankfurt Institute for Advanced Studies, Frankfurt am Main, Germany
[4]Xidian-FIAS international Joint Research Center, Frankfurt am Main, Germany

`{schaumloeffel, roignoguera}@em.uni-frankfurt.de`
`{aubret, triesch}@fias.uni-frankfurt.de`

## Abstract

Humans acquire semantic object representations from egocentric visual streams with minimal supervision, but the underlying mechanisms remain unclear. Importantly, the visual system only processes the center of its field of view with high resolution and it learns similar representations for visual inputs occurring close in time. This emphasizes slowly changing information around gaze locations. This study investigates the role of central vision and slowness learning in the formation of semantic object representations from human-like visual experience. We simulate five months of human-like visual experience using the Ego4D dataset and a state-of-the-art gaze prediction model. We extract image crops around predicted gaze locations to train a time-contrastive Self-Supervised Learning model. Our results show that exploiting temporal slowness when learning from central visual field experience improves the encoding of different facets of object semantics. Specifically, focusing on central vision strengthens the extraction of foreground object features, while considering temporal slowness, especially in conjunction with eye movements, allows the model to encode broader semantic information about objects. These findings provide new insights into the mechanisms by which humans may develop semantic object representations from natural visual experience. Code is available at https://github.com/t9s9/central-vision-ssl.

## 1 Introduction

Humans develop strong semantic object representations from an egocentric visual stream with only little supervision. These semantic representations reflect different non-perceptual facets of an object, such as its identity, fine-grained category, basic category or its typical context of occurrence. Models trained with self-supervised learning (SSL) are reasonably good models of biological vision (Zhuang et al., 2021), but they are poor models of visual learning, as they rely on different training data and learning mechanisms than humans. This may be the reason why they fail to model human semantic object similarity judgments (Mahner et al., 2025) and under-perform at recognizing objects when trained on visual experience similar to that of humans (Orhan, 2023; Orhan & Lake, 2024).

To learn semantic representations from their natural visual experience, humans may rely on two aspects of biological vision that are typically neglected in current models. First, the stimuli received by the human visual cortex differ structurally from egocentric videos. The human retina samples information from the center of the field of view more densely (Anstis, 1974; Wässle et al., 1989), such that high fidelity information is only available within several degrees from the center of the visual field, i.e., in central vision. As a consequence, central vision plays a crucial role in the formation of object-level representations in areas of the visual cortex related to semantic information (Quaia & Krauzlis, 2024; Yu et al., 2015). To compensate for the relatively low acuity in peripheral vision, humans move their gaze around 3 times per second to parse their environment. Second, a key principle

---

[*]Equal contribution
[†]Shared last authorship

of biological learning states that biological systems assign similar representations to close-in-time visual inputs (Miyashita, 1988). This may be important when learning from central visual experience. For instance, observing objects from different viewpoints may favor viewpoint-invariant object representations (Aubret et al., 2022a). In addition, consecutive scanning of objects within the same context may support object representations that encode their context of occurrence (Aubret et al., 2024a), a feature of human semantic object perception (Turini & Võ, 2022; Yeh et al., 2025). For instance, the presence of a knife often reflects a "kitchen" context. At present, it is unclear how self-supervision with a slowness objective and a human-like focus on central vision may shape the semantic properties of learned representations.

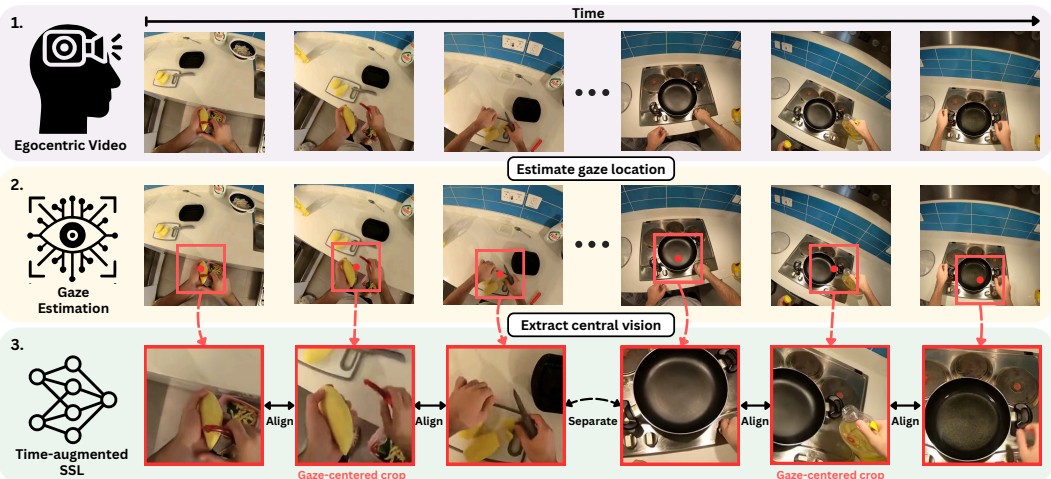

Figure 1: Illustration of our data generation and model training approach. 1) We extract frames from the egocentric dataset Ego4D (Grauman et al., 2022). 2) We predict the human gaze location (red dot) using a state-of-the-art model (Lai et al., 2024). 3) We train a time-augmented SSL model to align representations of gaze-centered crops (red rectangle) extracted from close-in-time frames.

In this paper, we investigate the combined role of emphasizing central vision and slowness learning for the formation of semantic object representations from human-like visual experience. We simulate 5 months of human-like visual experience using Ego4D (Grauman et al., 2022), a dataset that contains 3,670 hours of videos collected with head-mounted cameras. This dataset contains gaze locations on a subset of videos (45 hours). For the rest, we apply a state-of-the-art human gaze prediction model (Lai et al., 2024). To simulate the preferential sampling of the central visual field, we crop image regions centered on the measured/predicted gaze locations. Finally, we train a biologically inspired SSL model that learns visual representations which change slowly on the resulting visual sequence for one single epoch. Figure 1 summarizes our approach.

Our experiments demonstrate that learning slowly changing representations from central visual experience leads to more semantic object representations compared to a standard training on the whole field of view. Our analysis shows that this stems from a combined effect of emphasizing central vision, exploiting human eye movement patterns and the temporal slowness learning objective: while emphasizing central vision favors the extraction of foreground object information versus background information, learning with temporal slowness from human eye movements distills semantic information into the object representations. Overall, our work sheds light on how humans may build semantic object representations from natural visual experience. Furthermore, our approach may inspire more efficient learning strategies in embodied AI that produce more human-like semantic representations.

## 2 RELATED WORKS

**Egocentric SSL.** The increased availability of datasets collected with head-mounted cameras (Grauman et al., 2022; Sullivan et al., 2021; Long et al., 2024; Greene et al., 2024; Ma et al., 2024)

recently induced a surge for training egocentric SSL models (Orhan et al., 2024; He et al., 2022; Zhou et al., 2022; Emin Orhan, 2024). To the best of our knowledge, all these approaches use the entire high-resolution field of view captured by head-mounted cameras. Previous work found that endowing SSL models with representations that slowly change over time can slightly boost object learning (Orhan et al., 2020). A related line of work leverages egocentric data to train vision models useful for solving robotic tasks. VC-1 (Majumdar et al., 2023) is trained on egocentric and third-person videos. R3M (Nair et al., 2023) and VIP (Ma et al., 2023) both notably learn slowly changing representations on Ego4D. We show in Section 4 that training on gaze-based central vision elicits better object representations. Other works try to extract the correspondences between objects' views in videos to learn visual representations (Jabri et al., 2020; Venkataramanan et al., 2024; Salehi et al., 2023; Parthasarathy et al., 2023; Gordon et al., 2020). Here, we are rather interested in understanding how focusing on central vision may impact egocentric SSL.

**Time-based SSL.** Many previous works have proposed to learn similar representations for close-in-time visual inputs (Wiskott & Sejnowski, 2002; Földiák, 1991). More recently, this learning principle has been integrated into mainstream SSL methods (Schneider et al., 2021; Aubret et al., 2022a). However, these works do not leverage in-the-wild egocentric data, but synthetic data (Aubret et al., 2022a; Schaumlöffel et al., 2023) or curated visual sequences of interactions with objects (Aubret et al., 2024b; Sanyal et al., 2023; Aubret et al., 2024a). Other works use third-person videos (Sermanet et al., 2018), videos recorded by a car (Jayaraman & Grauman, 2015; 2016), movie video clips (Jayaraman & Grauman, 2016), the egocentric perspective of chicks (Pandey et al., 2024) or object-tracking datasets (Xu & Wang, 2021).

**Central and peripheral vision in deep learning.** Many studies have modeled the space-variant resolution of the retina in the context of artificial vision systems. Previous works have shown that this can make representations acquired via supervised learning more adversarially robust (Vuyyuru et al., 2020), improve the computational efficiency of the training process (Lukanov et al., 2021) and induce a stronger center bias (Deza & Konkle, 2020). Other works combined a bio-inspired focus on central vision with attention mechanisms for temporally extended image recognition (e.g. Almeida et al. (2018)). In the context of SSL, Wang et al. (2021a) argues that foveation can mimic the impact of the Crop/Resize data-augmentation, which is widespread in SSL. In line with our work, recent studies have combined retina modeling with time-based SSL. Aubret et al. (2022b) showed that a progressive blur towards the visual periphery can make visual representations slightly more transferable across backgrounds, and Yu et al. (2024) showed that gaze patterns in central vision may support the learning of view-invariant object representations. However, these two works trained SSL models with a tiny number of objects (10 and 24, respectively). In contrast, using human data at scale allows us to study the role of bio-inspired learning with respect to semantic recognition abilities.

**Learning context-wise object representations.** Only few works studied the emergence of similar visual representations for objects that co-occur in the same context. One work (Bonner & Epstein, 2021) proposed to learn similar representations for objects that co-occur in individual images of natural scenes. They did not study temporal relations between objects across time. The impact of such temporal relations on learning more semantic representations was studied by Aubret et al. (2024a) with a curated dataset showing egocentric rotations around images and hand-made statistics of object transitions. Thus, it remains unclear how and whether the natural experience of humans supports the construction of object representations reflecting their typical context of appearance.

## 3 METHOD

We aim to study the combined impact of high-resolution central vision and the slowness principle on visual learning in humans. We use the largest-to-date dataset of egocentric videos (Ego4D) and estimate human gaze locations with a state-of-the-art model of human gaze prediction (Section 3.1). To simulate the biological importance of central vision, we simply crop the visual area around a gaze location. To model biological learning, the created sequence of visual inputs feeds an SSL model that learns slowly changing representations, which is described in Section 3.2. Figure 1 illustrates the main steps of the pipeline. Finally, we evaluate the ability of our model to capture different semantic facets of objects, using the approaches detailed in Section 3.3 and Section 3.4.

## 3.1 DATASET

To simulate the visual experience of humans, we use the Ego4D dataset (Grauman et al., 2022). This dataset contains 3,600 hours of videos collected through head-mounted cameras, which corresponds to approximately 5 months of visual experience. This data was recorded by 931 participants coming from 74 worldwide locations wore a camera for one to ten hours. Thus, Ego4D arguably represents much more than 5 months of experience for a single average human in terms of diversity, although it is hard to make precise estimates. We use videos with a resolution of $540 \times 540$ pixels and extract their frames at approximately 5 fps, following previous findings that higher fps do not boost the learning process (Sheybani et al., 2024).

During frame extraction, we create small clips of 5 seconds (25 frames) that we sequentially load into memory. We gather 24 frames of these 25 frames and split them into three sequences of 8 frames. For Ego4D videos recorded with an eye-tracker (45 hours), we do not further process the frames and associate them with ground-truth gaze locations. For all other videos, we feed each sequence into GLC, a state-of-the-art model of human gaze prediction trained on the Ego4D subset that contains gaze locations (Lai et al., 2024). This model uses spatio-temporal information to generate a saliency map for each of the 8 frames. For each frame, we take the position of the most salient pixel as gaze location $(x_g, y_g)$. Even when the saliency map contains multiple peaks, the strongest peak is typically stable across adjacent frames, yielding a reliable and temporally consistent gaze trajectory. Compared to single-image saliency models (Riche & Mancas, 2016), processing multiple frames allows the model to generate a temporally consistent gaze location and leverage more cues (e.g. motion). Our final preprocessed dataset contains 64,380,024 images.

To simulate the importance of central vision in humans, we crop a $N \times N$ square region centered on the gaze location in each frame. If the crop extends beyond the image boundaries, we apply a minimal shift to keep it fully within the image.

## 3.2 BIO-INSPIRED LEARNING

Since most of the human visual experience is unsupervised, we train SSL models on the simulated experience in central vision. These models learn high-level visual representations without any explicit supervision, like human-provided labels. In this work, we focus specifically on the third version of Momentum Contrast (MoCoV3) (Chen et al., 2021), which is one of the best SSL models in the literature. The original MoCoV3 works by learning invariant representations to color- and spatial-based transformations of an image (e.g. horizontal flip, color jittering . . . ). To implement the biological principle of temporal slowness, we further adapt the model to also learn slowly changing visual representations, following (Aubret et al., 2022a; Pandey et al., 2024).

For a given input image $x_t$ in a batch, we randomly sample an indirect temporal neighbor $x_{t'}$ within a temporal window $\Delta T$, from the same video recording. The two images capture the same scene at different moments in time, providing a temporally varied view. We compute the embeddings of images $q_t = f_q(x_t)$ and $k_{t'} = f_k(x'_t)$ using a query feature extractor $f_q$ and a momentum feature extractor $f_k$, both implemented as neural networks. Finally, for a pair $(q_t, k_{t'})$, the query encoder is updated by minimizing the InfoNCE loss (van den Oord et al., 2019):

$$\mathcal{L}_{q_t} = -\log \frac{\exp\big(\mathrm{sim}(q_t, k_{t'})/\tau\big)}{\sum_{i=0}^{K} \exp\big(\mathrm{sim}(q_t, k_i)/\tau\big)} \tag{1}$$

where sim denotes cosine similarity, $\tau$ is a temperature hyperparameter, and $K$ represents the outputs of $f_k$ from the same training batch. Intuitively, the objective increases the similarity between representations of temporally close views ($x_t$ and $x_{t'}$) while enhancing the dissimilarity between all views ($x_t$ and $x_i$). The momentum encoder parameters $\theta_k$ are updated via an exponential moving average of the query encoder $\theta_q$: $\theta_k \leftarrow m\theta_k + (1-m)\theta_q$, with momentum coefficient $m$.

## 3.3 EVALUATION OF IMAGE RECOGNITION ABILITIES

We follow standard SSL transfer protocols, evaluating frozen representations via linear probing across diverse downstream tasks grouped by semantic focus (see below). For each dataset, we train a linear classifier on top of the frozen features of the pre-trained encoder for 100 epochs. We apply the

standard crop/resize and horizontal flip augmentations during training and report the accuracy on a center crop of validation images.

**Object categorization:** To assess the categorization ability of the models, we consider the ImageNet-1k (Russakovsky et al., 2015), ImageNet100 (Tian et al., 2020) and CIFAR100 (Krizhevsky et al., 2009) datasets, including reduced, balanced subsets of ImageNet-1k (1%, 10%) (Chen et al., 2020). We also analyze object categorization tasks that contain a tiny number of classes in Appendix D.3.

**Fine-grained object categorization:** Most classes in the two previous groups are for basic-level category recognition (e.g. car, trucks, bananas ...). Here, we instead assess categorization at the subordinate level (e.g. for cars, differentiating a 2012 VW Polo from a 2012 BMW M3). This requires a model to extract finer details about an object. We consider a wide range of subordinate categories: Flowers101 (Nilsback & Zisserman, 2008), Stanford Cars (Krause et al., 2013), Oxford Pet (Parkhi et al., 2012), FGVC-Aircraft (Maji et al., 2013), DTD (Cimpoi et al., 2014).

**Instance-level object recognition:** We evaluate object instance recognition when exposed in front of different backgrounds with varying orientations. We use ToyBox (Wang et al., 2017), COIL100 (Nene et al., 1996), Core50 (Lomonaco & Maltoni, 2017). Core50 mostly allows us to assess the robustness of the representation to changing backgrounds, while ToyBox and COIL100 present objects in different positions and orientations. We explain in Appendix B how we split the training and testing splits. We do not apply a center crop on COIL100.

**Scene recognition:** For scene recognition, we focus on Places365-standard (Zhou et al., 2017a). This dataset contains 1.8 million images from 365 scene categories and is commonly used to probe scene-level representations.

## 3.4 EVALUATION OF THE CONTEXT-WISE ORGANIZATION OF OBJECT REPRESENTATIONS

In order to evaluate whether the knowledge about 3D object co-occurrences can naturally emerge from our models, we compare the representations of our models with representations specifically built to encode objects' co-occurrence structure.

**Object Co-occurrence Representations:** To model the latent semantic structure of natural scenes, we extract object co-occurrence statistics from three large-scale image datasets: COCO (Lin et al., 2014), ADE20K (Zhou et al., 2017b), and Visual Genome (VG) (Krishna et al., 2017). These datasets vary in label density and semantic granularity. COCO contains coarse object categories with dense instance annotations; ADE has finer-grained, segmentation-level labels; and VG offers a rich, albeit noisy, semantic graph structure. For each dataset, we construct a co-occurrence matrix $X \in \mathbb{R}^{N_o \times N_o}$, where $N_o$ is the number of object classes and $X_{ij}$ counts how often the object $i$ appears with the object $j$ in the same image. We train GloVe (Global Vectors for Word Representation) (Pennington et al., 2014) on these matrices to derive low-dimensional representations that encode this co-occurrence structure. We refer to Appendix A for more details.

**Model-to-Semantics Alignment:** To assess whether neural network representations encode a semantic structure similar to the co-occurrence embeddings, we perform a representation similarity analysis using Centered Kernel Alignment (CKA) (Kornblith et al., 2019).

We map object classes from the co-occurrence matrices to their corresponding WordNet synsets (Miller, 1995). Then, we retrieve representative images from the THINGS dataset (Hebart et al., 2023), which contains isolated object instances with a naturalistic appearance. For each object, we extract activations from all layers of a given model and average across object images, resulting in a single feature vector per object and layer. We then compute the linear CKA score between each layer's object representation matrix and the GloVe embedding matrix. Alternatively, we concatenate the representations across layers to compute a global CKA score, which serves as a summary measure of semantic alignment for the entire model. We study the robustness of the vocabulary mapping and the selection of object images in Appendix D.7.

We repeat all evaluations across 100 GloVe seeds. We perform a paired t-test across the seeds to compare the CKA scores of each model under identical co-occurrence conditions and assess statistical significance between models. Throughout the experiment section, we report mean scores, standard deviations, and significance levels.

## 3.5 IMPLEMENTATION DETAILS

We adapt the *solo-learn* training pipeline and implementation of the MoCoV3 model (da Costa et al., 2022), with ResNet-50 (He et al., 2016) and ViT-B/16 (Dosovitskiy et al., 2020) backbones. For the MoCoV3 loss, a two-layer MLP (hidden dimension 4096) projects features into a 256-dimensional embedding space. Models are trained for one epoch on Ego4D; training longer yielded only minor gains ($\approx +0.5\%$) at substantial computational cost, due to the large but redundant dataset sampled at 5 fps. Full hyperparameters are given in Appendix C.

## 4 EXPERIMENTS

We aim to assess the impact of bio-inspired learning emphasizing central vision and harnessing a temporal slowness objective on the resulting visual representations. To this end, we compare our model "Bio-inspired Learning" to a baseline from previous work, "Frames Learning", which uses the full field of view and omits the slowness objective during training (Orhan & Lake, 2024).

## 4.1 BIO-INSPIRED LEARNING FROM NATURAL EXPERIENCE BOOSTS OBJECT RECOGNITION.

Table 1: Linear probe accuracy on various datasets across two architectures, grouped by semantic category. For each semantic group, we report the average recognition accuracy. For bio-inspired vision, we use $\Delta T = 3$s for ResNet50 and $\Delta T = 1$s for ViT.

| Dataset | ResNet50 | | ViT-B/16 | |
|---|---|---|---|---|
| | Frames Learning | Bio-inspired Learning | Frames Learning | Bio-inspired Learning |
| *Category recognition* | | | | |
| ImgNet-1k | 49.50 | **49.58** | 49.47 | **49.86** |
| ImgNet-1k 10% | **35.53** | 35.34 | 37.65 | **38.10** |
| ImgNet-1k 1% | 19.23 | **20.25** | 19.51 | **20.10** |
| ImgNet-100 | **70.44** | 70.34 | 70.04 | **70.12** |
| CIFAR100 | 53.53 | **59.21** | 61.73 | **62.67** |
| Average | 45.65 | **46.94** | 47.68 | **48.17** |
| *Fine-grained recognition* | | | | |
| DTD | 47.24 | **57.06** | 59.89 | **62.23** |
| FGVCAircraft | 12.83 | **15.77** | **28.87** | 28.60 |
| Flowers102 | 43.72 | **49.01** | 76.35 | **77.05** |
| OxfordIIITPet | 46.68 | **47.03** | 54.41 | **56.26** |
| StanfordCars | 18.70 | **23.25** | **33.30** | 33.26 |
| Average | 33.84 | **38.42** | 50.56 | **51.58** |
| *Instance recognition* | | | | |
| ToyBox | 89.75 | **92.61** | 92.94 | **95.03** |
| COIL100 | 64.53 | **80.12** | 79.24 | **86.94** |
| Core50 | 22.82 | **28.26** | **24.02** | 23.77 |
| Average | 59.03 | **67.00** | 65.40 | **68.58** |
| *Scene recognition* | | | | |
| Places365 | **43.02** | 42.95 | **44.49** | 39.84 |

We investigate whether bio-inspired learning improves object recognition abilities by comparing against SSL models learning without a temporal slowness objective and from raw frames. In Table 1, we observe that bio-inspired learning promotes category recognition, fine-grained recognition and object instance recognition. The improvement is particularly strong for fine-grained and object instance recognition. For scene recognition, training on full frames consistently leads to better results. We conclude that bio-inspired learning from a natural visual experience promotes better object recognition, but comes at the cost of impaired scene recognition. In Appendix D.1, we show that these results generalize to additional model architectures.

**Emphasizing Central vision makes representations more object-centered.** To investigate why bio-inspired learning impacts scene and object recognition differently, we first study the impact

of the size of the gaze-based crops $N$ on learning. In Figure 2, we observe a sweet spot for the intermediate crop sizes $N = 224$ and $N = 336$ for all object-centered datasets. This sweet spot is located at $N = 336$ for category and instance recognition, while $N = 224$ seems to be better for fine-grained recognition. $N = 112$ gives the worst semantic recognition accuracies, indicating that it probably dismisses too much information about the image. Interestingly, the scene recognition accuracy consistently increases as we enlarge the crop size.

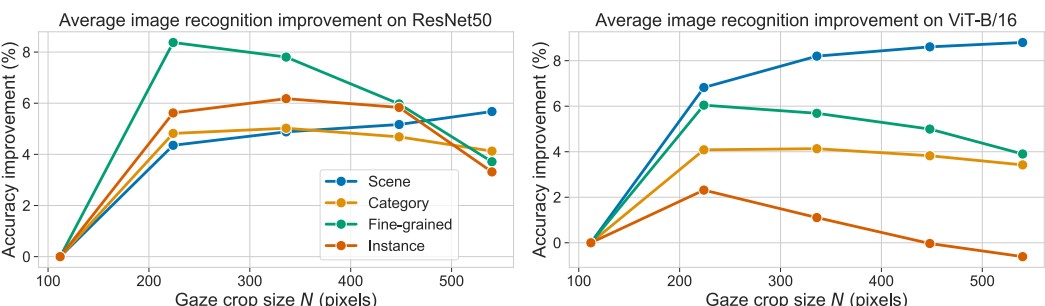

Figure 2: Impact of the gaze-based crop size on different groups of semantic image recognition tasks for ResNet50 and ViT-B/16. We compute the average improvement relative to $N = 112$. We use a temporal window of $\Delta T = 3$s. We omit error bars as variance across datasets is largely driven by differences in dataset difficulty rather than model instability; individual results are provided in the Appendix Table 9.

It could be that expanding the field of view mostly incites the model to extract more background features; background features may be more important for recognizing scenes than objects. To investigate this, we take the ImageNet-9 dataset, a dataset of natural images designed to investigate the background sensitivity of models (Xiao et al., 2021). The dataset provides different versions of the images, without background and with different ways to substitute the most salient object with background information. We compute the category recognition accuracy of our model with a linear probe trained on ImageNet-1k. We first find that training on central vision also benefits category recognition on normal images for this dataset (80% versus 75% on ResNet50). Then, we compute the recognition accuracy when removing the background (Missing background) and when removing the object (Missing object). When removing the object, we average the recognition accuracies of the different ways of removing the foreground object (cf. Xiao et al. (2021)). To obtain a measure of background and object sensitivity irrespective of the raw performance of the model, we subtract the recognition accuracy on normal images from those for the missing object and missing background images.

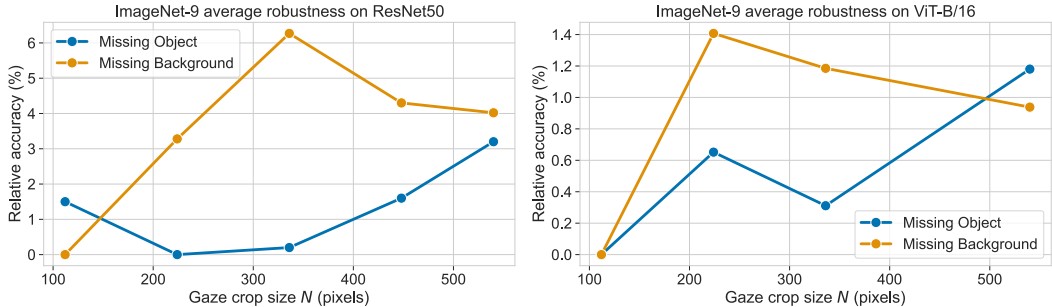

Figure 3: ImageNet-9 recognition sensitivity to missing background or missing foreground object. We show the relative improvement with respect to the worst model for the two settings. The higher, the more relatively robust is the representation to missing backgrounds or missing objects. We use a temporal window $\Delta T = 3$s.

In Figure 3, we clearly observe that training on an intermediate size of central vision ($N \in \{224, 336\}$) allows the model to rely more on the foreground object (missing background), and less on the

background (missing object). We assume this is because training on input from the central visual field removes much of the background information while often keeping the foreground object intact. For $N = 112$, there is an opposite trend, presumably because objects are often strongly cropped. Overall, we conclude that prioritizing central visual input leads to better object-centered representations because it reduces the amount of extracted background information.

**Slowness objective supports object learning.** Previous work on self-supervised learning from standard videos suggests that learning representations with a slowness objective can be beneficial if the representations of images that are around $t = 1$ second apart are made more similar (Xu & Wang, 2021). However, focusing on gaze-based central vision during egocentric learning provides semantically different temporal dynamics compared to using the whole field of view of, e.g., a movie clip. In Figure 4, we present the impact of the level of temporal slowness on visual representations acquired from such natural egocentric input. We observe that temporal slowness is critical for learning representations with respect to all the semantic aspects investigated ($\Delta T = 0$ versus $\Delta T = 3$ for ResNet50 and $\Delta T = 1$ for ViT). We note an exception for category recognition with ViT-B/16, for which the reason is currently unclear to us. We provide detailed results in Appendix E, which further show that the best temporal window $\Delta T$ is overall consistent for datasets within a group of semantic tasks.

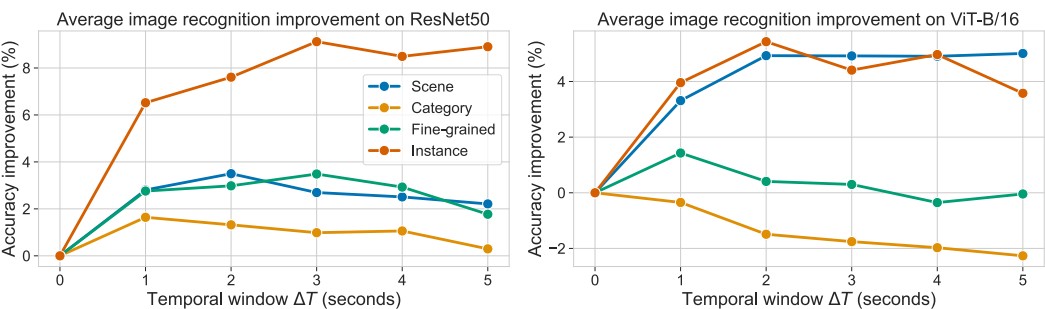

Figure 4: Impact of the temporal window of slowness learning on different groups of semantic image recognition tasks. We compute the average improvement for each group of tasks with respect to $\Delta T = 0$ seconds. We use a crop of size $N = 224$. We omit error bars as variance across datasets is largely driven by differences in dataset difficulty rather than model instability; individual results are provided in the Appendix Table 10.

## 4.2 HUMAN EYE MOVEMENT PATTERNS SUPPORT THE LEARNING OF OBJECT SEMANTICS

Humans actively look at salient objects inside a scene, raising the question of whether these specific eye movements are important for visual learning (Yu et al., 2024). To test this, we compare our bio-inspired model to baselines that ignore eye movements. The first baseline always crops the center of the frame, while the second uses fixation points predicted by a classical saliency model (Itti et al., 1998). Note that the resulting visual sequences still vary over time due to head movements of the camera wearer. In Table 2, we observe that bio-inspired training with estimated human gaze (GC) outperforms both center crops and saliency-based crops in most cases. Notably, for instance recognition, gaze-crops yield substantial improvements of +4.17% for ResNet and +1.14% for ViT over CC, while saliency-based crops close only a small fraction of this gap. The difference is even more pronounced for fine-grained and category recognition. Together with additional results in Appendix D.1 for ConvNeXt-B, these findings indicate that using simulated gaze locations of humans (weakly) boosts semantic object learning.

Human gaze behaviors are characterized by relatively long fixations interleaved by short saccadic eye movements, a pattern captured by the gaze estimation model (cf. Appendix D.4). To systematically investigate the relative importance of fixations versus saccades for object learning, we segment the data into alternations of likely fixations and saccades using a simple velocity threshold. In particular, we define a fixation as a consecutive sequence of frames during which the velocity of the gaze location in the image remains less than $P/200 \, \text{px} \, \text{ms}^{-1}$ , where $P$ is a parameter that denotes the maximum allowed gaze movement in $200 \, \text{ms}$. Then, during training, we ensure that two image frames forming

Table 2: Average recognition accuracies within semantic groups when pre-training with center-crop (*Center*), gaze-based visual cropping (*Gaze*) and saliency-based cropping (*Saliency*). We use a crop of size $N = 224$ and $\Delta T = 3s$.

| | ResNet50 | | | ViT-B/16 | |
| Semantic Group | Center | Saliency | Gaze | Center | Gaze |
|---|---|---|---|---|---|
| Category recognition | 46.58 | 45.36 | **46.94** | 46.53 | **46.76** |
| Fine-grained recognition | 37.77 | 36.28 | **38.42** | 50.31 | **50.35** |
| Instance recognition | 62.83 | 65.92 | **67.00** | 67.99 | **69.03** |

a positive pair during contrastive learning, even if spaced in time, belong to the same fixation. We train the model for different movement thresholds $P \in \{5, 15, 30, 45\}$, chosen to expose the model to varying proportions of fixations vs. saccades, following the statistics reported in Appendix D.4. In Figure 5, we observe that the model can indeed learn better object representations with respect to all object recognition abilities, when the fastest gaze movements are removed from the training sequence (i.e., when $P < \infty$). Under normal conditions, humans are aware of whether their eye movements are fixational or saccadic (O'Regan & Noë, 2001). Our results suggest that explicitly leveraging this distinction, *i.e.*, suppressing the learning signal during saccades, may enhance the emergence of semantic object representation.

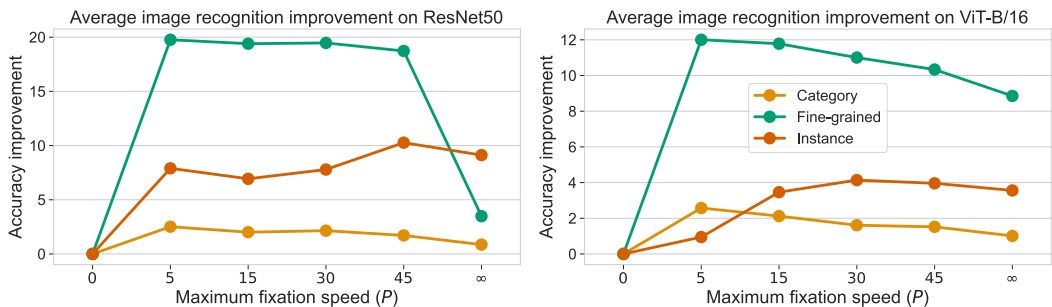

Figure 5: Impact of varying the gaze velocity threshold for identifying likely fixations. We report the average improvement for each group of semantic tasks relative to the $P = 0$ baseline, which corresponds to $\Delta T = 0$. All models are trained under the same settings as the bio-inspired models described in section 4.1.

## 4.3 SLOWNESS LEARNING BETTER ASSOCIATES OBJECTS THAT OCCUR IN THE SAME CONTEXT

In this section, we explore how slowness learning and central vision jointly shape the semantic similarity between categories of objects. In Table 3, bio-inspired learning yields object representations whose inter-object similarities more closely align with object co-occurrence statistics, capturing contextual relationships more effectively than "Frames learning". This effect is moderate with ResNet50 (+0.010 CKA score), but large with ViT (+0.028 CKA score). These improvements correspond to large effect sizes, indicated by Cohen's $d = 2.5$ for ResNet50 and $d = 7.0$ for ViT-B. We further observe that models trained with slowness learning produce significantly higher CKA similarity with co-occurrence embeddings than static models. This suggests that temporal slowness is essential for semantic learning.

In practice, the context-based organization of objects may be captured by learning both spatial and temporal co-occurrences. The former may be particularly pronounced when the image contains several objects, such as in the full frame. The latter may result from slowness learning when consecutively observing different objects (Aubret et al., 2024a). Thus, we investigate their relative role in shaping context-based object representations. In Table 3, we observe an increase in CKA similarity as we remove the focus on central vision (w/o Central vision), presumably because co-occurring objects are often spatially distributed and may not appear within the crop. However, we observe a minor

gap between "Bio-inspired Learning" and "without Slowness" suggesting that the integration of central vision over time partly compensates for the limited field of view. These findings generalize across different datasets (Appendix D.2). Overall, we conclude that slowness learning promotes the formation of context-aware object representations.

Table 3: CKA similarity between learned representations and GloVe-based object co-occurrence embeddings, computed on the COCO dataset. Higher values indicate stronger semantic alignment. w/o Slowness and w/o Central Vision correspond to the models labeled $\Delta T = 0$ in Figure 4 and $N = 540$ in Figure 2, respectively.

|  | Frames Learning | Bio-inspired Learning | w/o Slowness | w/o Central Vision |
|---|---|---|---|---|
| ResNet50 | $0.315 \pm 0.004$ | $\mathbf{0.325 \pm 0.004}$ | $0.320 \pm 0.004$ | $0.335 \pm 0.003$ |
| ViT-B/16 | $0.453 \pm 0.004$ | $\mathbf{0.481 \pm 0.004}$ | $0.406 \pm 0.004$ | $0.487 \pm 0.003$ |

## 5 CONCLUSION

Humans remain far more efficient in learning semantic visual representations than current machine learning systems. For example, the accuracy of the Top-5 linear probe with ImageNet-1k $1\%$ barely goes beyond $40\%$, compared to about $90\%$ for humans (Russakovsky et al., 2015; Orhan, 2023). Here, we investigated whether the relative importance of central vision (vs. peripheral vision) and the biological learning principle of slowness jointly support the learning of more semantic object representations from human-like visual experience. To this end, we simulated humans' gaze locations on the largest-to-date dataset of egocentric videos and extracted image regions surrounding the gaze locations. Then, we trained a biologically inspired SSL model that learns slowly changing visual representations on these data. Our extensive experiments demonstrate that extracting slowly changing information from the central visual field allows visual representations to better encode different facets of human object semantics. This includes the between-object similarity based on their context of co-occurrences, their basic category, fine-grained (or subordinate) category and their instance identity. Our analysis shows that emphasizing central vision elicits the extraction of more object-related features than background features. In addition, we found that fixational eye movements support such bio-inspired learning.

Our work has several limitations. First, the visuo-motor experience of infants during early development differs from the experience of adults modeled in the present work (Ayzenberg & Behrmann, 2024). Future work will have to investigate whether the conclusions drawn here generalize to visual experience modeled after that of young infants. Second, using a gaze estimation model sidesteps the question of how the developing visual system learns when and where to move its eyes. A complete model of visual development requires also modeling the learning of eye movement control strategies. Third, our model could be improved by incorporating more realistic retinal processing (Wang et al., 2021a), with a more gradual attenuation of the sampling of visual information towards the periphery. Yet, our work shows that prioritizing slow information from the central visual field permits the learning of strong semantic representations, marking a step toward a better understanding of the emergence of semantic object representations during human development.

### ACKNOWLEDGMENTS

This work was funded by the Deutsche Forschungsgemeinschaft: DFG project 5368 ("Abstract REpresentations in Neural Architectures (ARENA)") and DFG project 539642788, RO 6458/5-1 ("Learning from the Environment Through the Eyes of Children (LEECHI)"). This work was also supported by the Deutsche Forschungsgemeinschaft (German Research Foundation, DFG) under Germany's Excellence Strategy (EXC 3066/1 "The Adaptive Mind", Project No. 533717223). The authors acknowledge the ANR – FRANCE (French National Research Agency) for its financial support of the MeSMERise project ANR-23-CE23-0021-01. The authors gratefully acknowledge the computing time provided to them at the NHR Center NHR@SW at Goethe University Frankfurt (project autolearn). This is funded by the Federal Ministry of Education and Research, and the state governments participating on the basis of the resolutions of the GWK for national high performance

computing at universities (www.nhr-verein.de/unsere-partner). This work was also supported in part by computational resources provided by the MaSC high-performance computing cluster at Philipps-Universität Marburg, the Hessian Center For Artificial Intelligence and GENCI-IDRIS (Grant 2026-AD011017150).

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
