# A APPENDIX

## A GLOVE REPRESENTATIONS

GloVe (Pennington et al., 2014) trains low-dimensional embeddings from co-occurrence counts $X_{ij}$, by learning separate embeddings $\mathbf{v}_i$ and $\tilde{\mathbf{v}}_j$, and minimizing the objective:

$$J = \sum_{i,j=1}^{N} f(X_{ij}) \cdot \left( \mathbf{v}_i^\top \tilde{\mathbf{v}}_j + b_i + \tilde{b}_j - \log X_{ij} \right)^2, \quad (2)$$

where $f(x) = \min\left(1, (x/x_{\max})^\alpha\right)$ is a weighting function that downweights extremely frequent pairs, and $b_i$ and $\tilde{b}_j$ are learned biases.

However, visual co-occurrence is inherently symmetric. Unlike language, the joint appearance of objects $i$ and $j$ is bidirectional and unordered. To reflect this difference, we follow Gupta et al. (2019) and modify the GloVe objective by tying the embeddings and biases, resulting in $\mathbf{v}_i = \tilde{\mathbf{v}}_i$ and $b_i = \tilde{b}_i$.

For each dataset, we compute separate co-occurrence matrices for the training and test splits. We validate the GloVe hyperparameters by correlating the dot products of the learned embeddings with the log co-occurrence counts on the test set. The final configuration uses 128-dimensional symmetric embeddings with $x_{\max}$ set to the 0.9 quantile of the observed counts and $\alpha = 0.75$. To estimate representational variance, we train 100 GloVe models with different seeds for each dataset. Figure 6 illustrates GloVe vectors inferred from the COCO dataset, visualized with PaCMAP Wang et al. (2021b).

## B DATASETS

In Table 4, we present the datasets and benchmarks used in our experiments. The provided training splits are utilized for training the linear probe, and evaluations are conducted on the test splits. If a test split is unavailable, we use the validation split instead. For Core50, following the approach of Orhan & Lake (2024), we use 7 backgrounds for training and 5 backgrounds for testing. The task is relatively simple in COIL100, so we train a linear probe on only one image per class.

Table 4: Overview of datasets, including the number of datapoints in each split, the splits used for training a linear probe, and the splits used for evaluation.

| Dataset | Train Split | Test Split | Citation |
|---|---|---|---|
| *Category recognition* | | | |
| ImgNet-1k | 1,281,167 (train) | 50,000 (test) | (Russakovsky et al., 2015) |
| ImgNet-1k 10% | 128,116 (train) | 50,000 (test) | (Chen et al., 2020) |
| ImgNet-1k 1% | 12,811 (train) | 50,000 (test) | (Chen et al., 2020) |
| ImgNet-100 | 126,689 (train) | 50,000 (test) | (Tian et al., 2020) |
| CIFAR100 | 50,000 (train) | 10,000 (test) | (Krizhevsky et al., 2009) |
| *Fine-Grained Recognition* | | | |
| DTD | 1,880 (train) | 1,880 (test) | (Cimpoi et al., 2014) |
| FGVC-Aircraft | 3,334 (train) | 3,333 (test) | (Maji et al., 2013) |
| Flowers102 | 1,020 (train) | 6,149 (test) | (Nilsback & Zisserman, 2008) |
| OxfordIIITPet | 3,680 (trainval) | 3,669 (test) | (Parkhi et al., 2012) |
| StanfordCars | 8,144 (train) | 8,041 (test) | (Krause et al., 2013) |
| *Instance Recognition* | | | |
| ToyBox | 36,540 (train) | 15,660 (test) | (Wang et al., 2017) |
| COIL100 | 100 (train) | 7,100 (test) | (Nene et al., 1996) |
| Core50 | 90,000 (train) | 75,000 (test) | (Lomonaco & Maltoni, 2017) |
| *Scene Recognition* | | | |
| Places365 | 1,803,460 (train) | 36,500 (test) | (Zhou et al., 2017a) |

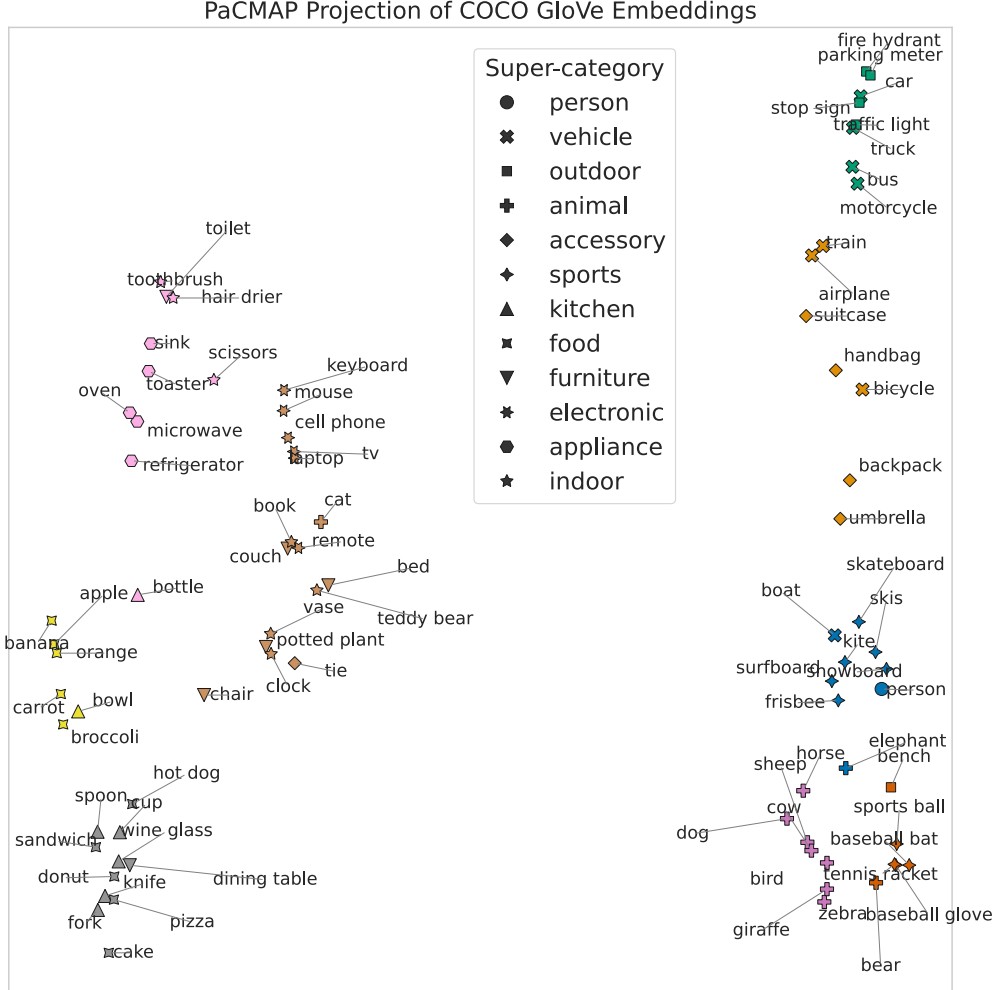

Figure 6: PaCMAP projection of GloVe embeddings learned from object co-occurrence statistics in the COCO dataset. Each point represents an object category. The marker shapes indicate COCO supercategories and colors reflect groupings obtained via a hierarchical clustering. Two broad clusters emerge, separating indoor (left) and outdoor objects (right). Within these clusters, the embeddings capture fine-grained spatial co-occurrence structures. For instance, a cat appears near a couch despite belonging to different supercategories (animal vs. furniture), reflecting their frequent co-occurrence in indoor scenes.

## C  IMPLEMENTATION DETAILS

We use different optimization strategies tailored to the different network architectures. For ResNet50, we employ the LARS optimizer (You et al., 2017) with a cosine decay learning rate schedule, without restarts, and a warm-up period set to 1% of the total training steps. The learning rate follows the linear scaling rule (Goyal et al., 2017), computed as $lr_{\text{base}} \times B/256$, with $lr_{base} = 1.6$ and a total batch size of $B = 512$. We apply a weight decay of $1e - 6$, maintain an exponential moving average (EMA) parameter of $m = 0.996$, and set the InfoNCE loss temperature to $\tau = 0.1$. Training runs in full precision across 8 GPUs, with a batch size of 64 per GPU. For ViT, we switch to the AdamW optimizer and adjust hyperparameters accordingly. ViT uses a base learning rate of $1.5e - 4$, weight decay of $0.1$, a batch size of 128 per device, and a warm-up period of 40% of training steps, with EMA increasing from $\tau = 0.99$ to $1.0$.

## D    ADDITIONAL RESULTS

### D.1    ADDITIONAL MODELS VALIDATE THE IMPORTANCE OF BIO-INSPIRED LEARNING

We ran our evaluation against HMAX (Riesenhuber & Poggio, 1999), a classic non-learnable vision model widely used in computational neuroscience. We observed that HMAX does not form object representations that are similarly "semantic" as those of our bio-inspired SSL models. On the ImageNet-100 linear probe, HMAX achieves $13.7\%$ accuracy, a drop of $56.6\%$ compared to our bio-inspired ResNet. Likewise, the semantic alignment to co-occurrences drops by $0.08$ as measured by the CKA similarity score.

To further test the generality of our findings, we ran additional experiments with ConvNeXt-B, a modern convolutional architecture that combines the design principles of CNNs and Transformers. In Table 5, the results show that our approach yields even larger improvements than those observed with ResNet50 and ViT-B/16 for object recognition.

Table 5: Linear probe accuracy on various datasets across two architectures, grouped by semantic category. For each semantic group, we report the average recognition accuracy. For bio-inspired vision, we use $\Delta T = 3$. For Frames Learning, we use $\Delta T = 0$

| | MocoV3 / ConvNeXt-B | | |
| --- | --- | --- | --- |
| Semantic Group | Frames Learning | Center Crop | Bio-inspired Learning |
| Category recognition | 34.30 | 41.24 | **42.96** |
| Fine-grained recognition | 28.39 | 46.03 | **46.09** |
| Instance recognition | 50.19 | **60.04** | 64.04 |
| Scene recognition | 39.64 | 35.42 | **41.26** |

### D.2    ARCHITECTURAL BIASES SHAPE THE CONTEXT-WISE ORGANIZATION OF OBJECT REPRESENTATIONS

In Section 4.1, we found that Transformer-based models (ViTs) outperform convolutional architectures (ResNets) across all spatial and temporal configurations. This difference may be due to fundamental architectural differences. ViTs use global self-attention to integrate spatial information throughout the input crop. In contrast, CNNs rely on localized receptive fields and hierarchical convolution, which may limit their ability to fully exploit the distributed context.

To further investigate these differences, we perform a layer-wise CKA analysis (see Figure 7), which compares the internal representations at each stage of the network with the co-occurrence embeddings. We observed that semantic alignment increases progressively across layers in ViT, peaking in the penultimate transformer block. In contrast, ResNet exhibits lower overall alignment, though we observe increasing alignment throughout its stages. Notably, ViT shows strong alignment in the mid-layers (layers 6–8), whereas ResNet requires deeper layers to reach a comparable level of alignment. This results is consistent with previous work showing that Transformers excel in capturing long-range dependencies more efficiently and benefit more from distributed contextual signals (Raghu et al., 2021).

In Figure 8 we provide extended results using co-occurrence matrices derived from the ADE (Zhou et al., 2017b) and Visual Genome (Krishna et al., 2017) datasets. These results replicate the patterns observed on COCO, which further validates the reported patterns across diverse semantic environments.

### D.3    10-CLASSES OBJECT CATEGORIZATION IS IRRELEVANT FOR ASSESSING OBJECT RECOGNITION

We also evaluate in Table 6 the models on easier categorization tasks that contain only 10 classes, namely STL10 (Coates et al., 2011) and CIFAR10 (Krizhevsky et al., 2009). We do not apply center crop on CIFAR10 and STL10. Interestingly, we observe inconsistent results between different visual backbones. Previous works found that models achieve high CIFAR10 accuracy by solely relying on background colors (Chiu et al., 2023). To further investigate this issue, we visualize with

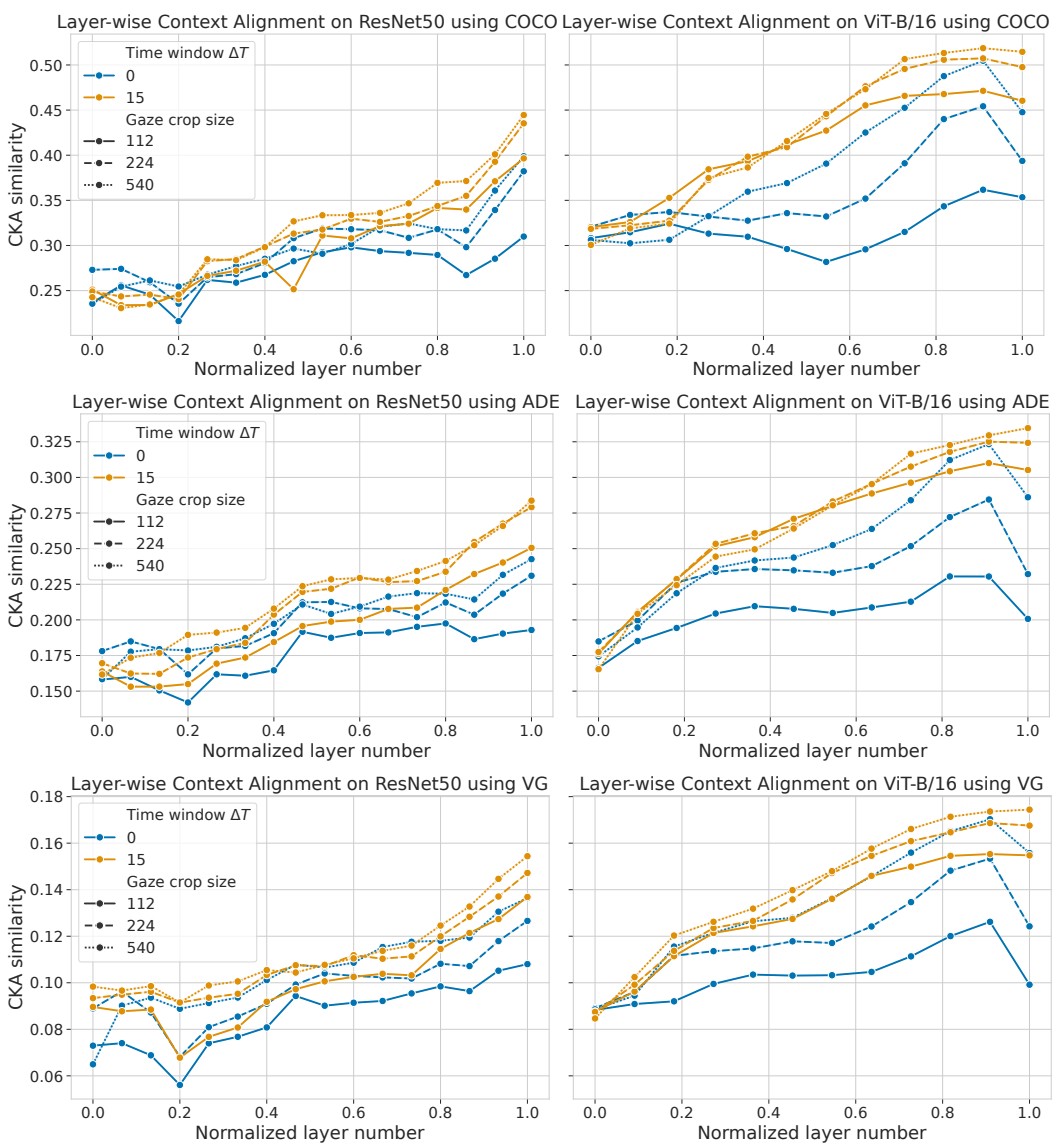

Figure 7: CKA similarity between representations from our trained model across the network hierarchy and GloVe-based object co-occurrence embeddings for the COCO, ADE, and Visual Genome (VG) datasets. Higher values indicate stronger semantic alignment. For ResNet, we evaluate the output after each residual block; for the Vision Transformer, after each transformer layer. Standard deviations, which are very small, are indicated in the plot.

STL10/ImgNet-1K (Figure 9) how the number of classes (10/1000) impacts the features attended by a linear probe. Spurious features (grass, sky) often suffice to classify objects in natural images (car, airplane) among ten classes (STL10). In contrast, more relevant object features (wheels, airplane wing) are necessary for 1000-way (Imgnet) classification. Thus, the low number of classes in STL10 and CIFAR10 make them inadequate datasets for evaluating object representations.

### D.4 GAZE MODEL HAS SIMILAR STATISTICS TO HUMANS

Here, we further analyze the gaze patterns. We present in Figure 10 (Left) the distribution of gaze centers across the entire training dataset of Ego4D. The gaze prediction model is biased toward the center, which may be a consequence of the head often following the eye movements in humans (Nakashima & Shioiri, 2014). However, we also find that the distribution has a non-null standard

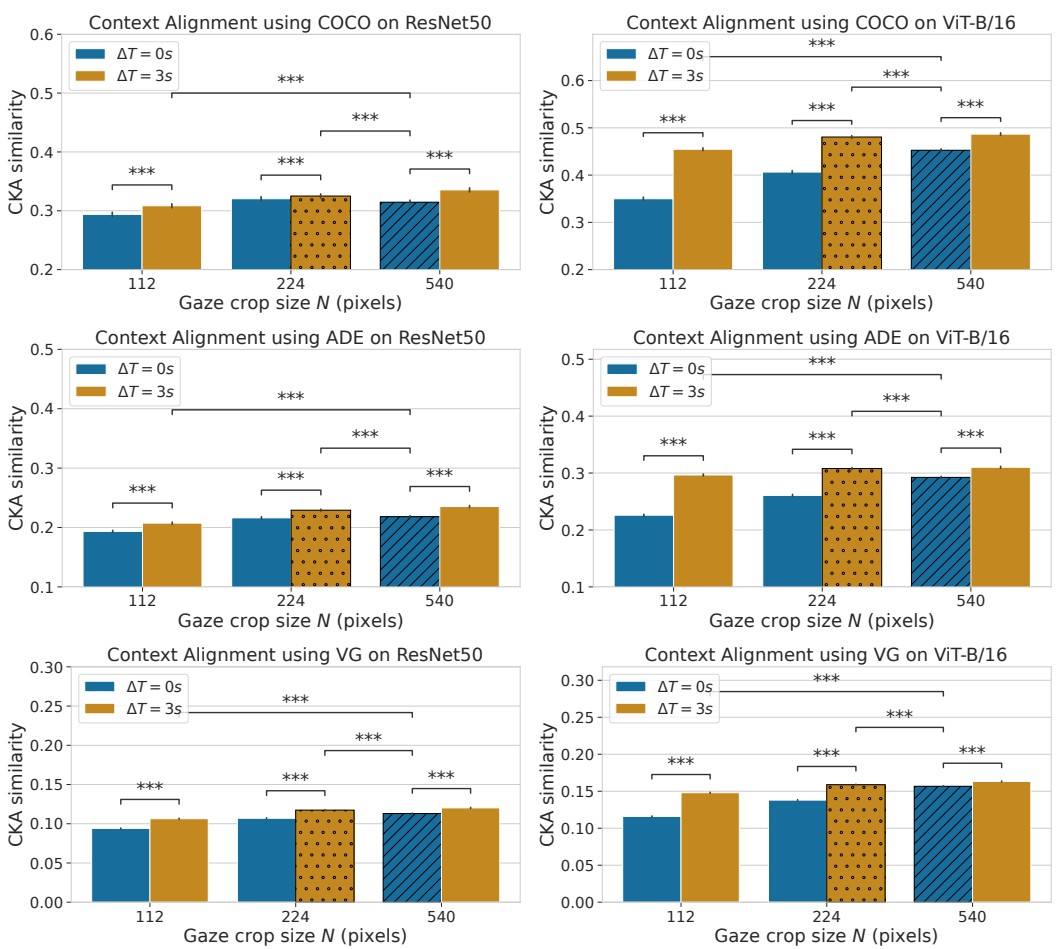

Figure 8: Further results for CKA similarity between learned model representations and GloVe-based object co-occurrence embeddings, computed on the COCO (also see table 3), ADE and Visual Genome (VG) dataset. Higher values indicate stronger semantic alignment. Each bar reflects the mean across multiple GloVe seeds; error bars denote standard deviation. Statistical significance between model configurations is indicated by asterisks ($*** : p < 0.001$). The dotted bar represents bio-inspired, and the dashed bar frames learning.

Table 6: Linear probe accuracy on categorization datasets with only 10 classes

| Model | Dataset | Full Frame | Central vision |
|---|---|---|---|
| ResNet50 | STL10 | **71.69** | 71.19 |
| | CIFAR10 | **79.57** | 79.32 |
| | Average | **75.63** | 75.26 |
| ViT-B/16 | STL10 | 78.39 | **79.19** |
| | CIFAR10 | 81.56 | **81.94** |
| | Average | 79.97 | **80.56** |

deviation, scattering around its mean location. This suggests that the gaze model may also capture subtle semantic information in the image. To further analyze eye movement statistics, we show the probability distribution of eye displacement magnitudes between samples separated by $200\,\mathrm{ms}$. in Figure 10 (Right). We observe that the probability distribution follows a classical exponential distribution, as found in humans (Schütt et al., 2019). Overall, these statistics further validate the gaze estimation model.

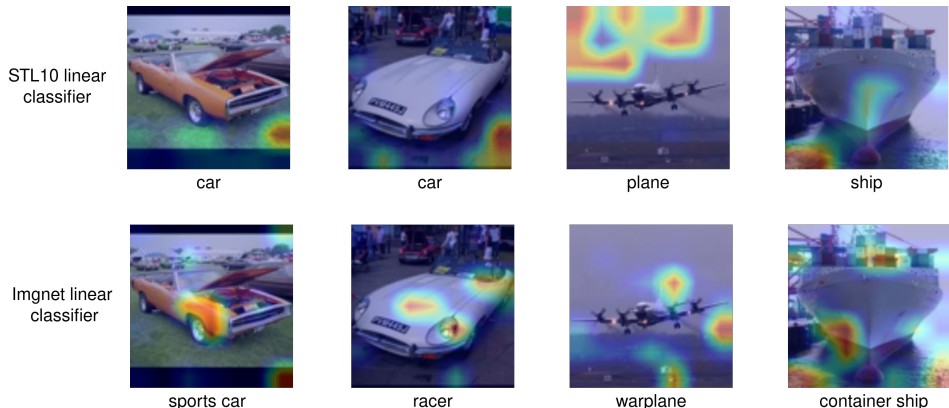

Figure 9: GradCam applied on a "Full frame" model ($\Delta T = 3$, $N = 224$) with two linear probes trained on STL10 and ImageNet-1k (100%). We use STL10 images. We also display models' class predictions, which we also used to apply GradCam.

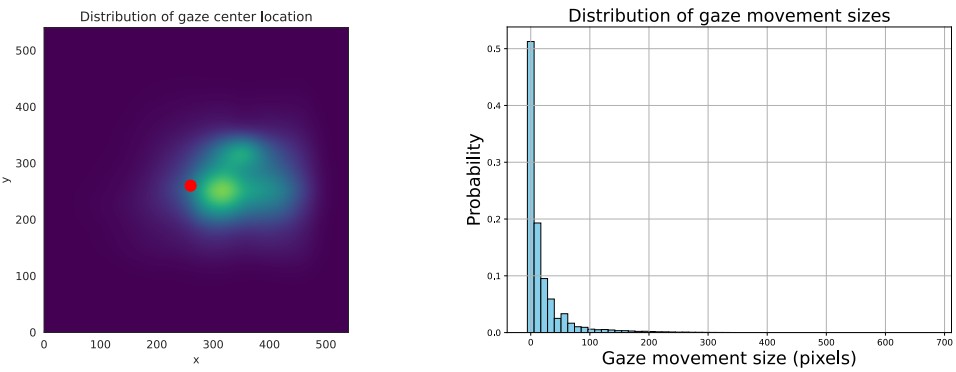

Figure 10: Left: Distribution of the gaze center location over the Ego4D dataset. The red dot symbolizes the center of the frame. Right: Probability distribution of the size of gaze movements.

### D.5 THE MODELS CAPTURE THE OBJECT CO-OCCURRENCES IN EGO4D

Here, we first verify that co-occurrences of objects in Ego4D reflect their context of occurrence. We select a set of concepts by computing the synsets of object concepts present in both the COCO and the Things datasets. We remove concepts that do not clearly belong to a context ("background", "person", "backpack", "cat"), leading to a set of concepts that belong to 10 contexts: "Urban streets" (bicycle, car, stop sign etc.), "Countryside" (horse, sheep, cow), "Savanna" (elephant, zebra, giraffe), "Sea" (kite, surfboard, boat), "Snow mountain" (skis, snowboard), "Sport field" (sports ball, baseball, baseball glove), "kitchen" (oven, wine glass, carrot etc.), "living room" (couch, television, potted plant), "bedroom" (teddy bear, bed, book etc.) and "bathroom" (sink, toilet, hair drier etc.).

To analyze the object co-occurrences in Ego4D, we utilize dense narrations that accompany short video clips. These narrations are human-annotated and provide written descriptions summarizing actions (e.g. "picks a green sponge from the sink"). From those, we extracted the co-occurrence matrix of selected synsets of nouns as a proxy for object co-occurrences within each short clip. In Figure 11A, we observe coherent clusters such as kitchen contexts (cup, fork) and street scenes (car, motorcycle), indicating that Ego4D provides meaningful contextual statistics.

Furthermore, we want to verify whether our bio-inspired model captures the co-occurrences of Ego4D. In Figure 11B), we compute the cosine similarity matrix between representations of Things images that belong to the selected concepts. In practice, we randomly sample 10 images per concept and average the pairwise similarity for each pair of concepts. As a representation, we choose the first ReLU activation of the MoCoV3 (ViT-B/16) projection layer. A qualitative analysis of Figure 11

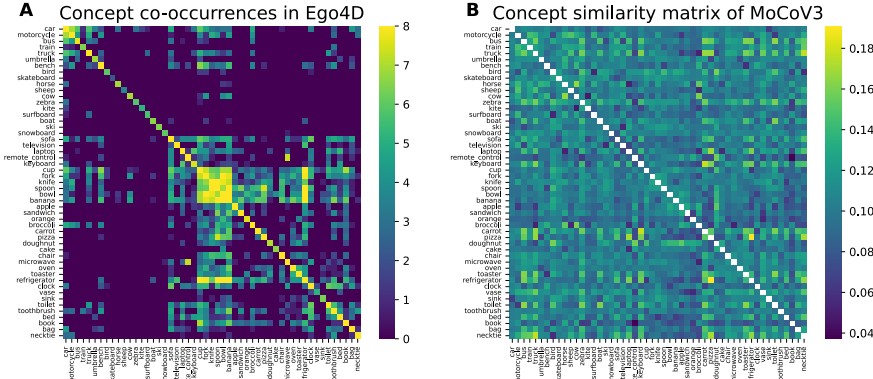

Figure 11: A) Logarithmic number of co-occurrences extracted from Ego4D for pairs of COCO/Things concepts. We clip the maximum values to 8 for visibility. B) Cosine similarity matrix between concepts with MoCoV3 and ViT-B/16. We use the first ReLU activation of the projection head. We mask the diagonal (intra-category images similarity) for visualization purpose.

shows that the model captures some key context-wise co-occurrences (*e.g.* pizza and refrigerator, or truck and motorcycle) but fails to capture important ones (*e.g.* cup, fork, banana). Although many semantic and perceptual dimensions also drive human similarity judgments (Mahner et al., 2025), this indicates that there is a margin for improvement.

### D.6 EVALUATION OF THE GAZE MODEL

For completeness, we report the validation results of the gaze prediction model, as evaluated in the original study Lai et al. (2024). The standard approach to evaluate a gaze model is to estimate the amount of overlap between the saliency map generated by the model and a saliency map derived from the ground-truth gaze location. Especially, the F1 score is the harmonic mean between Precision and Recall. Precision quantifies how many predicted salient pixels are true fixations, and Recall quantifies how many ground-truth fixations are recovered by the prediction. GLC notably achieves a F1 score of 43.1 on Ego4D, establishing the method as a reasonable approximation of human's gaze location. We refer to the original paper for a qualitative evaluation of the method.

To evaluate whether gaze-prediction quality meaningfully affects our method, we compare models trained on ground-truth gaze with models trained on predicted gaze. Since the gaze predictor (Lai et al., 2024) is trained on the ground-truth subset, we cannot reapply it to the same data. Furthermore, the test split used to evaluate the gaze model in the original study contains only 7 videos (Xiao et al., 2025), which is insufficient to train a reasonable vision model. Instead, we construct three random Ego4D subsets with matched semantic diversity, following the entropy-based diversity measure of Xiao et al. (2025). We train identical ResNet50 MoCoV3-TT models on the ground-truth dataset and on the three predicted-gaze datasets, matching the number of training steps across all runs.

As shown in Table 7, models trained with true human gaze perform only slightly better than those trained with predicted gaze. For category and fine-grained recognition, the gaps lie within the standard deviation, and for instance recognition the difference remains small relative to overall variability. Since the datasets cannot be matched perfectly in content, part of this gap likely reflects diversity differences rather than gaze-estimation accuracy. Overall, the results indicate that the output of the gaze model is a reasonable approximation of the true gaze for training our vision models.

### D.7 ROBUSTNESS OF THE MAPPING PIPELINE FOR CKA ANALYSIS

Here we assess the robustness of the mapping pipeline used in our CKA analysis, which aims to quantify the context-wise structure of model representations. Our semantic analyses rely on a two-stage mapping pipeline: (i) mapping object labels from the co-occurrence datasets (COCO, ADE20K, VG) to WordNet synsets, and (ii) mapping these synsets to THINGS object categories for extracting model activations (see Section 3.4). The second stage is determined by the THINGS dataset, which

Table 7: Comparison of linear-probe accuracy across semantic categories for a ResNet50 (MoCo v3) trained on ground-truth gaze versus our predicted gaze. For predicted gaze, we report the mean and standard deviation across three subsets and show the performance difference relative to ground truth.

| Semantic Group | Ground-Truth Gaze | Predicted Gaze | Difference |
|---|---|---|---|
| Category recognition | 32.08 | $30.68 \pm 2.18$ | 1.40 |
| Fine-grained recognition | 37.45 | $36.97 \pm 2.76$ | 0.48 |
| Instance recognition | 56.44 | $54.67 \pm 1.47$ | 1.78 |

provides standardized category labels derived from WordNet synsets. An additional potential source of variability arises from the selection of exemplar images in THINGS that represent each mapped category.

To evaluate the robustness of this mapping pipeline, we performed two analyses. First, we restricted the evaluation to co-occurrence classes with an unambiguous, one-to-one mapping to WordNet synsets (65 of 80 classes; e.g., *spoon*, *plate*, *chair*). We manually excluded categories such as *potted plant* or *stop sign*, which are ambiguous because their visual or semantic definitions are either inconsistent in THINGS (e.g., *potted plant* covers many plant species and pot types) or have multiple possible interpretations (e.g., *stop sign* varies in shape, design, and contextual presentation). Second, we repeated the full analysis while randomly sampling half of the available THINGS exemplars per synset across 10 independent runs, thereby probing the sensitivity to exemplar selection.

As shown in Table 8, across both robustness checks we observed small variations in absolute CKA values, as expected, but the relative ordering of models and all key conclusions remained unchanged. These results confirm that our findings are robust to reasonable variations in both stages of the vocabulary–mapping pipeline.

Table 8: Robustness ablation of CKA similarity between learned representations and GloVe-based object co-occurrence embeddings, computed on the COCO dataset. Higher values indicate stronger semantic alignment. w/o Slowness and w/o Central Vision correspond to the models labeled $\Delta T = 0$ in Figure 4 and N=540 in Figure 2, respectively.

| | Frames Learning | Bio-inspired Learning | w/o Slowness | w/o Central Vision |
|---|---|---|---|---|
| | *THINGS exemplar subsampling* | | | |
| ResNet50 | $0.331 \pm 0.005$ | $\mathbf{0.340 \pm 0.005}$ | $0.336 \pm 0.005$ | $0.351 \pm 0.005$ |
| ViT-B/16 | $0.471 \pm 0.005$ | $\mathbf{0.500 \pm 0.005}$ | $0.424 \pm 0.005$ | $0.507 \pm 0.005$ |
| | *Unambiguous category subset* | | | |
| ResNet50 | $0.340 \pm 0.006$ | $\mathbf{0.348 \pm 0.006}$ | $0.344 \pm 0.006$ | $0.360 \pm 0.006$ |
| ViT-B/16 | $0.475 \pm 0.005$ | $\mathbf{0.503 \pm 0.004}$ | $0.428 \pm 0.005$ | $0.511 \pm 0.004$ |

# E  COMPLETE RESULTS DATA

We show in Table 9 and Table 10 the detailed results of Figure 2 and Figure 4, respectively. Our results are overall consistent in each group of semantic tasks.

Table 9: Detailed results of Figure 2. Top-1 accuracy on different datasets for various gaze sizes $N$ for a fixed time window $\Delta T = 3$s.

| Dataset | ResNet50 | | | | | ViT-B/16 | | | | |
|---|---|---|---|---|---|---|---|---|---|---|
| | N=112 | N=224 | N=336 | N=448 | N=540 | N=112 | N=224 | N=336 | N=448 | N=540 |
| *Category recognition* | | | | | | | | | | |
| ImgNet | 43.52 | 49.58 | **50.57** | 49.68 | 48.98 | 43.16 | 48.07 | **48.48** | 48.37 | 47.61 |
| ImgNet 10% | 29.44 | 35.34 | **36.28** | 35.65 | 35.27 | 31.47 | 36.10 | **36.22** | 35.69 | 34.56 |
| ImgNet 1% | 15.77 | 20.25 | **20.66** | 20.40 | 20.11 | 15.73 | **19.07** | 18.87 | 18.62 | 17.91 |
| ImgNet-100 | 63.45 | 70.34 | 71.46 | **71.48** | 70.90 | 63.50 | **69.06** | 68.72 | 69.04 | 68.42 |
| CIFAR100 | 58.44 | **59.21** | 56.76 | 56.83 | 55.99 | 59.53 | 61.51 | 61.76 | 60.79 | **62.00** |
| Average | 42.12 | 46.94 | **47.15** | 46.81 | 46.25 | 42.68 | 46.76 | **46.81** | 46.50 | 46.10 |
| *Fine-grained recognition* | | | | | | | | | | |
| DTD | 50.69 | **57.06** | 54.56 | 53.93 | 52.18 | 58.30 | **62.18** | 61.76 | 60.21 | 60.59 |
| FGVCAircraft | 9.89 | **15.77** | 14.81 | 14.63 | 14.42 | 21.58 | **27.61** | 26.74 | 26.83 | 25.09 |
| Flowers102 | 45.41 | **49.01** | 48.21 | 43.25 | 46.16 | 69.83 | **74.84** | 74.35 | 73.50 | 72.90 |
| OxfordIIITPet | 26.39 | 47.03 | **47.58** | 45.48 | 34.86 | 47.71 | **55.75** | 54.71 | 54.08 | 52.26 |
| StanfordCars | 17.86 | 23.25 | **24.11** | 22.82 | 21.19 | 24.11 | 31.39 | **32.42** | 31.87 | 30.19 |
| Average | 30.05 | **38.42** | 37.85 | 36.02 | 33.76 | 44.31 | **50.35** | 49.99 | 49.30 | 48.21 |
| *Instance recognition* | | | | | | | | | | |
| ToyBox | 89.78 | 92.61 | **92.74** | 92.15 | 92.59 | 94.22 | **95.14** | 95.14 | 95.04 | 94.79 |
| COIL100 | 76.53 | **80.12** | 79.49 | 79.35 | 75.56 | 86.44 | **87.46** | 86.35 | 85.66 | 84.76 |
| Core50 | 17.82 | 28.26 | **30.44** | 30.14 | 25.92 | 19.49 | **24.48** | 21.97 | 19.35 | 18.77 |
| Average | 61.38 | 67.00 | **67.56** | 67.21 | 64.69 | 66.71 | **69.03** | 67.82 | 66.68 | 66.11 |
| *Scene recognition* | | | | | | | | | | |
| Places365 | 38.60 | 42.95 | 43.48 | 43.77 | **44.27** | 34.63 | 41.45 | 42.83 | 43.24 | **43.43** |

Table 10: Detailed results of Figure 4. Top-1 accuracy on different datasets when training from different time windows $\Delta T$ and fixed crop size $N = 224$.

| $\Delta T$ Dataset | ResNet50 | | | | | | ViT-B/16 | | | | | |
|---|---|---|---|---|---|---|---|---|---|---|---|---|
| | 0 | 1 | 2 | 3 | 4 | 5 | 0 | 1 | 2 | 3 | 4 | 5 |
| *Category recognition* | | | | | | | | | | | | |
| ImgNet | 48.64 | **50.18** | 49.60 | 49.58 | 49.12 | 48.90 | **50.40** | 49.86 | 48.90 | 48.07 | 47.71 | 47.94 |
| ImgNet 10% | 34.53 | **35.98** | 35.62 | 35.34 | 35.12 | 34.77 | **38.93** | 38.10 | 36.67 | 36.10 | 35.91 | 35.65 |
| ImgNet 1% | 18.52 | **20.37** | 20.29 | 20.25 | 19.74 | 19.85 | 20.08 | **20.10** | 19.23 | 19.07 | 18.69 | 18.64 |
| ImgNet-100 | 69.08 | **71.26** | 70.94 | 70.34 | 70.90 | 70.86 | **70.54** | 70.12 | 69.44 | 69.06 | 69.08 | 67.98 |
| CIFAR100 | 59.02 | 60.20 | 59.94 | 59.21 | **60.21** | 56.89 | 62.64 | **62.67** | 60.89 | 61.51 | 61.33 | 61.06 |
| Average | 45.96 | **47.60** | 47.28 | 46.94 | 47.02 | 46.25 | **48.52** | 48.17 | 47.03 | 46.76 | 46.54 | 46.25 |
| *Fine-grained recognition* | | | | | | | | | | | | |
| DTD | 46.39 | 54.78 | 55.52 | **57.06** | 56.58 | 55.41 | 59.52 | **62.23** | 61.70 | 62.18 | 61.28 | 61.97 |
| FGVCAircraft | 14.75 | 14.21 | 14.93 | **15.77** | 14.84 | 13.64 | 27.82 | **28.60** | 26.95 | 27.61 | 26.68 | 27.01 |
| Flowers102 | 46.50 | 48.89 | 47.74 | 49.01 | **49.04** | 47.32 | 76.63 | **77.05** | 75.70 | 74.84 | 75.15 | 74.37 |
| OxfordIIITPet | 46.35 | 48.20 | **48.34** | 47.03 | 45.86 | 44.50 | 53.40 | **56.26** | 54.87 | 55.75 | 54.49 | 55.23 |
| StanfordCars | 20.71 | 22.41 | 23.08 | **23.25** | 23.02 | 22.68 | 32.89 | **33.26** | 33.08 | 31.39 | 30.90 | 31.47 |
| Average | 34.94 | 37.70 | 37.92 | **38.42** | 37.87 | 36.71 | 50.05 | **51.48** | 50.46 | 50.35 | 49.70 | 50.01 |
| *Instance recognition* | | | | | | | | | | | | |
| ToyBox | 86.99 | 90.99 | 92.29 | **92.61** | 92.56 | 92.52 | 92.08 | 95.03 | 95.22 | 95.14 | 95.17 | **95.44** |
| COIL100 | 69.67 | 78.36 | 80.05 | 80.12 | **82.43** | 79.79 | 78.83 | 86.94 | 87.59 | 87.46 | **88.90** | 86.56 |
| Core50 | 16.98 | 23.84 | 24.12 | **28.26** | 24.12 | 28.04 | 22.95 | 23.77 | **27.34** | 24.48 | 24.69 | 22.59 |
| Average | 57.88 | 64.40 | 65.49 | **67.00** | 66.37 | 66.78 | 64.62 | 68.58 | **70.05** | 69.03 | 69.59 | 68.20 |
| *Scene recognition* | | | | | | | | | | | | |
| Places365 | 40.26 | 43.06 | **43.76** | 42.95 | 42.77 | 42.47 | 36.53 | 39.84 | 41.46 | **41.87** | 41.44 | 41.54 |