# OpenReview forum: "Temporal Slowness in Central Vision Drives Semantic Object Learning"
_ICLR.cc/2026/Conference — ICLR 2026 Poster_

### Official Review · Reviewer_mtJa · 2025-10-21

**Soundness:** 2
**Presentation:** 3
**Contribution:** 3
**Rating:** 6
**Confidence:** 4

**Summary:**

This paper investigates the role of central vision and slowness learning in semantic object learning. Specifically, it simulates the human-like visual experience using ego videos. The experimental results show that the combination of temporal slowness and central vision can improve the encoding of different semantic facets of object representations. Experiments are conducted on various tasks, including object categorization, Fine-grained object categorization, instance-level object recognition, and scene recognition, to validate the image recognition capabilities of the proposed approach.

**Strengths:**

1. The authors conducted extensive experiments to validate the effectiveness of the proposed bio-inspired learning strategy.

2. The utilization fo Ego4D dataset and gaze coordinates to mimic central vision is a kind of novel

3. This paper is well organized and easy to read.

**Weaknesses:**

1. The proposed bio-inspired learning is inferior to frames learning on scene recognition under both the ResNet and ViT backbones. This result is not well discussed.

2. The influence of gaze estimation is not thoroughly discussed. The authors leverage a state-of-the-art gaze estimation model to obtain cropped regions that mimic central vision. Would the accuracy of gaze estimation affect the performance of the proposed method? Furthermore, if the salient regions were cropped instead to construct the central vision data, would the method still perform well?

3. This paper contains some typos, such as a missing period in line 093.

**Questions:**

1. Would the accuracy of gaze estimation affect the performance of the proposed method? Furthermore, if the salient regions were cropped instead to construct the central vision data, would the method still perform well?

2. The proposed bio-inspired learning is inferior to frame-based learning on scene recognition under both the ResNet and ViT backbones. Could the authors provide a comprehensive discussion of this?

---

> ### Author Response · Authors · 2025-11-21
>
> We thank the reviewer for their detailed comments and address each weakness (W1–W3) and question (Q1–Q2) below.
>
> ---
>
> ### Q1/W2 — Influence of Gaze Estimation
> #### Effect of Gaze Estimation
> To evaluate whether gaze-prediction quality meaningfully affects our method, we compare models trained on ground-truth gaze with models trained on predicted gaze. Since the gaze predictor is trained on the ground-truth subset, we cannot reapply it to the same data. Furthermore, the test split used to evaluate the gaze model in the original study contains only 7 videos (Lai et al. 2024), which is insufficient to train a reasonable vision model. Instead, we construct three random Ego4D subsets with matched semantic diversity, following the entropy-based diversity measure of Xiao et al. (2025). We train identical ResNet50 MoCoV3-TT models on the ground-truth dataset and on the three predicted-gaze datasets, matching the number of training steps across all runs.
>
> As shown in Table R1, models trained with true human gaze perform only slightly better than those trained with predicted gaze. For category and fine-grained recognition, the gaps lie within the standard deviation, and for instance recognition the difference remains small relative to overall variability. Since the datasets cannot be matched perfectly in content, part of this gap likely reflects diversity differences rather than gaze-estimation accuracy. Overall, the results indicate that the gaze model is a reasonable approximator of the true gaze for training our vision models. We provide these results in the appendix D6 of the revised manuscript.
>
> Table R1: Performance comparison between ground-truth and predicted gaze
>
> | Semantic Group             |  Ground-Truth Gaze  |     Predicted Gaze     |  Difference  |
> |:---------------------------|:-------------------:|:----------------------:|:------------:|
> | Category recognition       |        32.08        |      30.68 ± 2.18      |     1.40     |
> | Fine-grained recognition   |        37.45        |      36.97 ± 2.76      |     0.48     |
> | Instance recognition       |        56.44        |      54.67 ± 1.47      |     1.78     |
>
> #### Using saliency-based crops instead.
> First, we would like to clarify that the gaze prediction model we use can be seen as a very advanced saliency model. Our pipeline extracts a single gaze point by taking the global maximum of the predicted heatmap (Section 3.1). This corresponds to selecting the most salient location as inferred by the model.
>
> We are currently training a ResNet50 with gaze locations generated by a classical saliency model based on only low-level visual features by Itti et al. (2002). Although the experiment is still running due to computational constraints, intermediate evaluations show that this model performs substantially worse than our bio-inspired version. This aligns with our findings in Section 4.2: saliency models typically generate temporally inconsistent fixation points, and such inconsistency negatively impacts slowness-based learning. We will report the final results in Table 2 and in a follow-up comment in the coming days.
>
> ### W1/Q2 — Scene Recognition vs. Object Recognition
> Thanks for raising this point. We discuss this effect in Section 4.1 (paragraphs 2–4). To summarize:
>
> Full-frame training retains the entire field of view and thus provides strong background and contextual signals, which dominate scene recognition tasks such as Places365. In contrast, central-vision sampling removes much of the background and therefore encourages object-centered representations. This shift is beneficial for tasks that depend on object identity (fine-grained and instance-level recognition), but naturally leads to lower performance on explicit scene-classification benchmarks. This effect is expected and aligns with our initial intuition.
>
> We have revised the section for clarity and improved the discussion beginning around line 310.
>
> ### W3 — Typos
> We have carefully reviewed the manuscript and corrected the noted typos, including the missing period on line 093.
>
> ---
>
> We thank the reviewer for the helpful feedback. We have incorporated all requested changes and believe the revisions substantially strengthen the paper. We hope the updated manuscript satisfactorily addresses the concerns raised.
>
> ### References:
> Xiao, J., Yan, B., Zhang, J., Wang, J., Li, C., Cheng, Z., & Zhai, G. (2025). Data Assessment for Embodied Intelligence. arXiv preprint arXiv:2511.09119.
>
> Itti, L., Koch, C., & Niebur, E. (2002). A model of saliency-based visual attention for rapid scene analysis. IEEE Transactions on pattern analysis and machine intelligence, 20(11), 1254-1259.

---

> > ### Comment · Area_Chair_wore · 2025-11-22
> >
> > Hi Reviewer,
> >
> > The authors have submitted their responses to your reviews. Please take a look and let the authors know if you have any further questions or concerns. Thank you again for your contributions to ICLR!
> >
> > Best regards,
> > AC

---

> > ### Comment · Reviewer_mtJa · 2025-11-23
> > **Response to Author Response**
> >
> > Dear Authors,
> > Thank you for your reply. My concerns have been resolved. I hope these discussion can be included in the revised version.
> > Best regards.

---

> > > ### Author Response · Authors · 2025-11-25
> > > **Follow-up on W2 / Q1 — Saliency-based Control**
> > >
> > > We thank the reviewer for confirming that all concerns have been resolved. We have incorporated the corresponding
> > > discussions into the revised manuscript and highlighted all changes in blue for clarity.
> > >
> > > ### Q1/W2 Results with saliency-based crops (missing results)
> > >
> > > The experiment using fixation points from a classical saliency model has now completed. As shown in Table R2, replacing
> > > gaze-based central vision with saliency-based crops consistently impairs object recognition across all semantic levels.
> > > This confirms the importance of using (estimated) humans' gaze locations for training our bio-inspired model. We have
> > > added these results to Section 4.2 of the manuscript.
> > >
> > > **Table R2: Bio-inspired vs. saliency-based central vision (ResNet50, MoCo-V3-TT)**
> > >
> > > | Semantic Group           | Bio-Inspired | Saliency-Based Crops | Difference |
> > > |--------------------------|-------------:|---------------------:|-----------:|
> > > | Category recognition     |        46.94 |                45.36 |   **1.58** |
> > > | Fine-grained recognition |        38.42 |                36.28 |   **2.14** |
> > > | Instance recognition     |        67.00 |                65.92 |   **1.08** |
> > > | Scene recognition        |        42.95 |                42.78 |   **0.17** |
> > >
> > > ---
> > > We appreciate the reviewer’s constructive feedback, which has helped improve the clarity and completeness of the
> > > submission. We hope the revisions fully address all remaining points, and we are happy to provide any further
> > > information if needed.

---

### Official Review · Reviewer_Qmz9 · 2025-10-27

**Soundness:** 1
**Presentation:** 2
**Contribution:** 1
**Rating:** 2
**Confidence:** 4

**Summary:**

This paper investigates the role of central vision and temporal slowness in the formation of semantic object representations in humans. It collects video data from Ego4D dataset and generates gaze coordinates with a existing gaze prediction model. It then crops image regions around gaze coordinates (central vision) and train a time-contrastive self-supervised learning model (temporal slowness).  The experiments assess the impact of learning visual representations with bio-inspired central vision and temporal slowness.

**Strengths:**

1) The paper aims to study the role of central vision and temporal slowness in the formation of semantic object representations in humans. The motivation is interesting.
2) Extensive experiments are conducted to evaluate effectiveness of the proposed feature learning on four downstream vision recognition tasks.

**Weaknesses:**

1) **Lack of technical novelty**. The technical approach of the paper, that is the extraction of gaze-centered image crops with existing gaze prediction model and the self-supervised constrastive learning guided by temporal distance, are very basic technical processes. Thus there is lack of sufficient technical contributions of the paper.
2) **Lack of sufficient performance comparison**.  The paper didn't make sufficient comparison with SOTA feature learning methods to validate the superior performance of the proposed method. There are not a few work each year focusing on self-supervised feature learning in the computer vision community, however, the paper only compares with a baseline of “Frames Learning”.

**Questions:**

1) Current gaze prediction methods based on video input are more like saliency prediction which produces heatmap for each image frame. How do you crop the image when such heatmap is distributed like multi-Gaussians?
2) Have you considered the cases of gaze transition in a video clip which causes dissimilarity of visual appearance between adjacent framews.

**Details Of Ethics Concerns:**

NA.

---

> ### Author Response · Authors · 2025-11-21
>
> We thank the reviewer for their constructive feedback and address the raised weaknesses (W1–W2) and questions (Q1–Q2)
> below.
>
> ---
>
> ### W1 — Clarification on the Objective and Contribution of the Paper
>
> Our objective is to investigate how semantic object representations can emerge from human-like egocentric visual
> experience. This is not to propose a new SOTA architecture or algorithmic novelty for computer vision. Our contribution
> lies in demonstrating, for the first time, that a biologically motivated slowness learning mechanism supports the
> emergence of more semantic object representations when applied to human everyday first person central visual field
> experience. To our knowledge, this has never been shown before and the components of our system (gaze prediction model,
> self-supervised learning based on slowness, emphasizing central vision) have never been combined before.
>
> ### W2 — On Comparison to SOTA feature learning
>
> Our goal was not to build a SOTA computer vision model. Instead, to support our findings, we compare only to our own
> controlled variants, ensuring consistency in data, architecture and hyperparameters. A comparison to SOTA vision systems
> would be neither fair nor aligned with the scientific question we aim to address.
>
> ### Q1 — Extraction of Gaze Locations from Heatmaps
>
> Thanks for raising this point. The gaze estimator produces a heatmap for each frame. We then convert this to a single
> gaze location by selecting the global maximum of the heatmap. Even when the heatmap contains multiple peaks (e.g.
> multi-Gaussian structures), the dominant peak typically remains stable across consecutive frames. This rule eliminates
> the need for additional heuristics and produces a consistent fixation trajectory. We improved the clarity of the
> description of our Methods (Section 3.1) in the revised version.
>
> ### Q2 — Effect of gaze transitions
>
> Yes, we explicitly study this phenomenon. In Section 4.2, we find that saccadic eye movements (gaze transitions)
> interfere with the learning of coarse categorization, fine-grained categorization and instance recognition. Preventing
> interference further boosts the object recognition abilities of the model (Figure 10). Regarding context-based
> representations, we plan to further disentangle how saccades harm or help in future work.
>
> ---
>
> We hope these clarifications address the reviewer’s concerns and help in reassessing the contribution of our work.

---

> > ### Comment · Area_Chair_wore · 2025-11-22
> >
> > Hi Reviewer,
> >
> > The authors have submitted their responses to your reviews. Please take a look and let the authors know if you have any further questions or concerns. Thank you again for your contributions to ICLR!
> >
> > Best regards,
> > AC

---

### Official Review · Reviewer_pvCu · 2025-10-30

**Soundness:** 3
**Presentation:** 3
**Contribution:** 3
**Rating:** 6
**Confidence:** 3

**Summary:**

The paper investigates whether combining central vision via gaze-centered crops with temporal slowness in self-supervised learning improves object-centric representation learning from long egocentric video. The pipeline predicts gaze where ground truth is unavailable, constructs gaze-centered crops, and trains a time-augmented SSL variant that aligns crops across nearby timesteps. Linear-probe evaluations indicate consistent gains for category, fine-grained, and instance recognition, while scene recognition can favor full-frame training. The analysis further explores context learning by comparing representation similarity to object co-occurrence embeddings.

**Strengths:**

- The integration of central vision with temporal slowness is a focused, biologically motivated idea that appears to yield practical improvements on object-centric tasks, suggesting value for egocentric and embodied learning communities.
- The empirical exploration includes reasonable sweeps over crop sizes and temporal pairing that reveal interpretable behavior and replicate across backbones.
- The end-to-end pipeline is presented clearly with intuitive figures that explain how gaze-centered cropping and temporal pairing interact.

**Weaknesses:**

- There is a heavy reliance on the predicted gaze rather than ground-truth gaze, and it is not accompanied by any calibration/error metrics. The paper could benefit from reporting prediction error characteristics and relating them to performance when ground-truth gaze is substituted for the eye-tracked subset.
- It remains unclear whether the gains arise from human-like fixations or from general object-biased views. The paper could benefit from controls using fixed center crops, saliency-only crops without temporal constraints, and motion-centric crops that ignore gaze, matched for compute and augmentations.
-  Important details such as batch sizes, optimizer and learning-rate schedules, EMA settings, total updates, and effective tokens/pixels are not consolidated, making it difficult to rule out under- or over-training effects. The paper could benefit from a unified table and training curves across methods.
- The method appears less favorable for scene recognition. The paper could benefit from experiments that blend full-frame and gaze-centered crops within batches or adjust crop scales during temporal pairing to probe whether object gains can be retained while narrowing the scene gap.
- Effect sizes, multiple-comparison adjustments, and potential vocabulary-mapping biases are not fully discussed. The paper could benefit from explicit reporting and robustness checks that vary the mapping pipeline.

**Questions:**

1. How does representation quality change on the eye-tracked subset when training with ground-truth gaze, predicted gaze, fixed center crops, and saliency-only crops? Providing accuracy deltas with confidence intervals would clarify the specific contribution of human-like fixation information.

2. What are the calibration and error properties of the large-scale gaze predictor (e.g., angular error, spatial bias, per-activity breakdown), and how do these correlate with downstream linear-probe metrics when substituting ground truth?

3. Were mixed training regimes attempted (for example, combining full-frame and gaze-centered crops within a batch or varying crop scale for temporally paired views), and how did these affect object-centric and scene-centric metrics?

4. How sensitive are results to the fixation threshold and grouping rules? Histograms of fixation/saccade durations and speeds, plus an ablation over the threshold, would make the dependence more transparent.

---

> ### Author Response · Authors · 2025-11-21
>
> We thank the reviewer for the detailed and constructive feedback. Below we address the raised weaknesses (W1–W5) and
> questions (Q1–Q4), and we indicate how we have clarified or extended the manuscript in response.
>
> ---
>
> ### Q1/W1/W2 — Evaluating the Contribution of Gaze Information
>
> In Section 4.2, we found that models trained on predicted gaze achieve slightly better performance than those trained on
> a center crop. We emphasize these results in our revision. Furthermore, we are currently training a
> ResNet50 with gaze locations generated by a classical Saliency model (Itti et al. 2002). Processing the saliency is very slow and we lack
> time to complete this experiment. However, preliminary evaluations of the model at intermediate training stages show
> that the model performs much worse. This is expected, as such a saliency model naturally produces inconsistent gaze
> positions over time: we found in Section 4.2 that this tends to impair the learning process. We will add the final
> results in Table 2, Section 4.2.
>
> To evaluate whether gaze-prediction quality meaningfully affects our method, we compare models trained on ground-truth
> gaze with models trained on predicted gaze. Since the gaze predictor is trained on the ground-truth subset, we cannot
> reapply it to the same data. Furthermore, the test split used to evaluate the gaze model in the original study contains
> only 7 videos (Lai et al. 2024), which is insufficient to train a reasonable vision model. Instead, we construct three
> random Ego4D subsets with matched semantic diversity, following the entropy-based diversity measure of Xiao et al. (
> 2025). We train identical ResNet50 MoCoV3-TT models on the ground-truth dataset and on the three predicted-gaze
> datasets, matching the number of training steps across all runs.
>
> As shown in Table R1, models trained with true human gaze perform only slightly better than those trained with predicted
> gaze. For category and fine-grained recognition, the gaps lie within one standard deviation, and for instance
> recognition the difference remains small relative to overall variability. Since the datasets cannot be matched perfectly
> in content, part of this gap likely reflects diversity differences rather than gaze-estimation accuracy. Overall, the
> results indicate that the gaze model is a reasonable approximator of the true gaze for training our vision models. We
> provide these results in the appendix D6 of the revised manuscript.
>
> Table R1: Performance comparison between ground-truth and predicted gaze
>
> | Semantic Group           | Ground-Truth Gaze | Predicted Gaze | Difference |
> |:-------------------------|:-----------------:|:--------------:|:----------:|
> | Category recognition     |       32.08       |  30.68 ± 2.18  |    1.40    |
> | Fine-grained recognition |       37.45       |  36.97 ± 2.76  |    0.48    |
> | Instance recognition     |       56.44       |  54.67 ± 1.47  |    1.78    |
>
> ### W3 — Missing Consolidated Training Details
>
> Due to space constraints, they are provided in Appendix C, including all the details to facilitate their rationale and
> reproducibility of results.
>
> ### W5 — Robustness of Mapping Pipeline
>
> We are not entirely sure to understand the reviewer’s comment. Please, could they elaborate on their concern about
> effect sizes and multiple-comparison adjustments ? Additionally, we would appreciate further explanations about what is
> meant by "potential vocubulary-mapping biases” and “mapping pipeline”, so that we can address these points
> appropriately.
>
> ### Q2 — Calibration and Error Properties of the Gaze Predictor
>
> In order to make our manuscript more self-contained, we report in the revised Appendix D.6 the quantitative evaluation
> of the gaze predictor provided in the original study (Lai et al., 2022). In our analyses, we also observe that the model
> exhibits a pronounced center bias (Figure 10, Appendix D.4). To our knowledge, there is no established method for
> systematically controlling the prediction errors of gaze models in order to examine their specific impact on downstream
> recognition performance. One possible direction would be to conduct a thorough comparison and analysis of existing gaze
> prediction models, but this lies beyond the scope of the present work.
>
> (continued below)

---

> > ### Author Response · Authors · 2025-11-21
> >
> > ### Q3/W4 — Mixed Training Regimes (Full-Frame + Gaze, Varying Crop Scales)
> >
> > Following reviewer’s suggestion, we trained a ResNet50 with crop size randomly sampled in {112,224,336,448,540}. In the
> > following table, we observe that varying the crop scale yields slightly worse recognition accuracies than our original
> > method, except for instance recognition. There is no consistent effect over object-centric metrics.
> >
> > | Semantic Group           | Bio-inspired learning | Varying crop scale |
> > |:-------------------------|:---------------------:|:------------------:|
> > | Category recognition     |        46.943         |       46.44        |
> > | Fine-grained recognition |        38.424         |       38.33        |
> > | Instance recognition     |         67.00         |       70.69        |
> > | Scene recognition        |         42.95         |       42.22        |
> >
> > ### Q4 — Sensitivity to Fixation Thresholds and Grouping Rules
> >
> > We provide in Figure 10 (Right) a histogram of gaze movement sizes. All bars left to a given fixation threshold
> > cumulatively quantify the number/time of fixations. In Figure 5, we show how varying the fixation threshold impacts our
> > results. To complement this analysis, we ran two additional experiments with Maximum Fixation Speed P = {5, 15}/200
> > px.ms-1, which we added to Figure 5. For both architectures, the best object recognition accuracies sit at P = 5/200
> > px.ms-1, except for instance recognition, which reaches a peak at P = {30, 45}/200 px.ms-1. Overall, this supports our
> > claim that saccadic eye movements interfere with temporal slowness for learning about objects.
> >
> > ---
> >
> > We thank the reviewer again for the careful and insightful analysis. The requested clarifications, reorganized
> > comparisons, and additional experiments have strengthened the presentation and interpretation of our results. We hope
> > the improved exposition helps the reviewer reassess the contribution of the work.
> >
> > ### References:
> >
> > Itti, L., Koch, C., & Niebur, E. (2002). A model of saliency-based visual attention for rapid scene analysis. IEEE
> > Transactions on pattern analysis and machine intelligence, 20(11), 1254-1259.
> >
> > Lai, B., Liu, M., Ryan, F., & Rehg, J. M. (2024). In the eye of transformer: Global–local correlation for egocentric
> > gaze estimation and beyond. International Journal of Computer Vision, 132(3), 854-871.
> >
> > Xiao, J., Yan, B., Zhang, J., Wang, J., Li, C., Cheng, Z., & Zhai, G. (2025). Data Assessment for Embodied Intelligence.
> > arXiv preprint arXiv:2511.09119.

---

> > > ### Comment · Area_Chair_wore · 2025-11-22
> > >
> > > Hi Reviewer,
> > >
> > > The authors have submitted their responses to your reviews. Please take a look and let the authors know if you have any further questions or concerns. Thank you again for your contributions to ICLR!
> > >
> > > Best regards,
> > > AC

---

> > > ### Comment · Reviewer_pvCu · 2025-11-25
> > >
> > > Thank you for addressing most of my concerns. I appreciate the new experiments comparing ground-truth and predicted gaze (Table R1) and the fixation-threshold ablation and histograms, which strengthen my confidence in the approach's robustness. The crop-scale experiment partially addresses my suggestion about mixed-training regimes.
> > >
> > > Two concerns remain somewhat open:
> > > - The mechanism itself: whether gains truly arise from human-like fixations vs generic object-biased crops. This still lacks a full set of controls (e.g., motion-only crops, finalized saliency-based baselines).
> > > - My earlier point about “vocabulary-mapping biases” referred to the way object categories/co-occurrence statistics are mapped into semantic embeddings for the analysis section. It would help to discuss effect sizes and how sensitive these analyses are to reasonable changes in that mapping pipeline.
> > >
> > > Clarifying these two would help readers better understand the paper.

---

> > > > ### Author Response · Authors · 2025-11-29
> > > > **Response to Remaining Concerns (Part 1)**
> > > >
> > > > ### 1. Do gains arise from human-like fixations or from generic object- or motion-biased crops?
> > > >
> > > > We appreciate the reviewer’s request for stronger controls. First, our experiments in Figure 4 varying the temporal
> > > > window $\Delta T$ show that models trained without temporal information ($\Delta T = 0$) tend to perform worse than those
> > > > using temporally extended inputs. As suggested, we now provide another complementary analysis that more directly
> > > > addresses whether the gains can be explained by generic saliency-based crops. We tested a classic saliency model (Itti
> > > > et al., 2002), which provides object-biased but frame-wise, temporally incoherent crops. As shown in the below table R2,
> > > > replacing the gaze prediction model with this basic saliency model reduces performance across all object-centric
> > > > semantic groups. Taken together, these analyses indicate that neither the temporal context provided by ego-motion nor
> > > > frame-wise saliency is sufficient to explain the gains observed with gaze-based cropping. Therefore, the temporal
> > > > structure captured by gaze-based cropping appears important for learning semantic object representations. We have added
> > > > these results to Section 4.2 of the manuscript.
> > > >
> > > > Second, we agree that it would be an interesting analysis to check whether the advantages of using humans’ gaze come
> > > > from humans’ attentional bias to external motion. However, in head-mounted egocentric video, the optic flow is dominated
> > > > by head movement, making motion-saliency ill-defined without explicit ego-motion compensation. Developing such a model
> > > > is nontrivial and we view it as promising future work. Importantly, this does not affect the claims made in the paper:
> > > > temporal slowness combined with human gaze patterns support semantic object learning.
> > > >
> > > > **Table R2: Bio-inspired vs. saliency-based central vision (ResNet50, MoCo-V3-TT)**
> > > >
> > > > | Semantic Group           | Gaze-based crops | Saliency-Based Crops | Difference |
> > > > |:-------------------------|:----------------:|:--------------------:|:----------:|
> > > > | Category recognition     |      46.94       |        45.36         |  **1.58**  |
> > > > | Fine-grained recognition |      38.42       |        36.28         |  **2.14**  |
> > > > | Instance recognition     |      67.00       |        65.92         |  **1.08**  |
> > > > | Scene recognition        |      42.95       |        42.78         |  **0.17**  |
> > > >
> > > > (continued below)

---

> > > > > ### Author Response · Authors · 2025-11-29
> > > > > **Response to Remaining Concerns (Part 2)**
> > > > >
> > > > > ### 2. Is the mapping pipeline robust?
> > > > >
> > > > > We thank the reviewer for the helpful clarification.
> > > > >
> > > > > Our analysis involves two mapping stages: (i) mapping labels from the co-occurrence datasets (COCO, ADE20K, VG) to
> > > > > WordNet synsets, and (ii) mapping these synsets to THINGS object categories for extracting model activations (Section
> > > > > 3.4). The second stage is directly defined by the THINGS dataset, which provides standardized category labels from
> > > > > synsets. We improved clarity of both stages in the manuscript. A further potential source of variability is the choice
> > > > > of exemplar images from THINGS used to represent each category.
> > > > >
> > > > > To assess the robustness of the overall mapping pipeline, we conducted two further checks which we now provide. First,
> > > > > we restricted the analysis to co-occurrence classes with an unambiguous one-to-one mapping to WordNet synsets (65/80
> > > > > classes; e.g., spoon, plate, chair), excluding ambiguous cases such as potted plant or stop sign. These categories are
> > > > > ambiguous because they are either represented inconsistently in THINGS (e.g., potted plant mixes many plant species and
> > > > > container types), or because they have multiple possible semantic or visual interpretations (e.g., stop sign can refer
> > > > > to different shapes, designs, or contextual appearances). Second, we repeated the analysis while randomly sampling half
> > > > > of the available THINGS exemplars per synset across 10 runs.
> > > > >
> > > > > In both robustness checks, absolute CKA values varied slightly, as expected, but the relative ordering of models and all
> > > > > main conclusions remained unchanged. We report the corresponding robustness analyses in the appendix D7. These results
> > > > > indicate that our findings are not sensitive to reasonable variations in either mapping stage.
> > > > >
> > > > > **Table R3: Robustness of semantic alignment to variations in the vocabulary–mapping pipeline**
> > > > >
> > > > > |                                   | Frames Learning | **Bio-inspired Learning** | w/o Slowness  | w/o Central Vision |
> > > > > |-----------------------------------|:---------------:|:-------------------------:|:-------------:|:------------------:|
> > > > > | **_THINGS exemplar subsampling_** |                 |                           |               |                    |
> > > > > | ResNet50                          |  0.331 ± 0.005  |     **0.340 ± 0.005**     | 0.336 ± 0.005 |   0.351 ± 0.005    |
> > > > > | ViT-B/16                          |  0.471 ± 0.005  |     **0.500 ± 0.005**     | 0.424 ± 0.005 |   0.507 ± 0.005    |
> > > > > | **_Unambiguous category subset_** |                 |                           |               |                    |
> > > > > | ResNet50                          |  0.340 ± 0.006  |     **0.348 ± 0.006**     | 0.344 ± 0.006 |   0.360 ± 0.006    |
> > > > > | ViT-B/16                          |  0.475 ± 0.005  |     **0.503 ± 0.004**     | 0.428 ± 0.005 |   0.511 ± 0.004    |
> > > > >
> > > > > **Regarding effect size.** Given the computational cost of training SSL models, and the well-established stability of their
> > > > > results at convergence, it is standard practice to train one SSL model per experiment (i.e. He et al. (2019)).
> > > > > Unfortunately, this does not allow us to systematically estimate standard deviation over accuracy metrics or effect
> > > > > size. Regarding differences of CKA between Frames learning and Bio-inspired learning, we now computed Cohen’s d. We
> > > > > obtain d = 2.5 and d = 7 for ResNet50 and ViT-B, respectively, which is indicative of big effects. This clarifies that
> > > > > our bio-inspired learning has a major effect on the learning of context-based representations and we added it into the
> > > > > revised paper (Section 4.3).
> > > > >
> > > > > ---
> > > > >
> > > > > We hope these clarifications address the reviewer’s concern and are happy to provide any further details where needed.
> > > > >
> > > > > ---
> > > > >
> > > > > ### References:
> > > > >
> > > > > Itti, L., Koch, C., & Niebur, E. (2002). A model of saliency-based visual attention for rapid scene analysis. IEEE
> > > > > Transactions on pattern analysis and machine intelligence, 20(11), 1254-1259.
> > > > >
> > > > > He, Kaiming et al. “Momentum Contrast for Unsupervised Visual Representation Learning.” 2020 IEEE/CVF Conference on
> > > > > Computer Vision and Pattern Recognition (CVPR) (2019): 9726-9735

---

### Official Review · Reviewer_KB1f · 2025-11-01

**Soundness:** 3
**Presentation:** 4
**Contribution:** 3
**Rating:** 6
**Confidence:** 5

**Summary:**

In this paper, the authors explore the contribution of central vision and slowness learning in visual representations of objects within the human brain via bio-inspired self-supervised learning. Using gaze locations as estimated by a gaze prediction model, the method extracts crops that resemble human central vision and train a time-augmented SSL model that aligns crops from the same short time segments within Ego4D recordings. Through several experiments on different granularities of object and scene recognition using both ResNet and VIT architectures, the authors present evidence for their claim that central vision and slowness learning has a strong effect on learning semantic object-level visual representations.

**Strengths:**

1. The paper is well-written and easy to follow.

2. The set of experiments and analyses done to present evidence of the claims is quite meticulous and impressive. I want to particularly appreciate the analysis on CKA similarity between learned representations and Glove-based object co-occurrence embeddings.

3. The performance metrics for some of the tasks like fine-grained and instance-level recognition are quite impressive.

**Weaknesses:**

1. Even though the paper focuses on central vision, simply ignoring peripheral vision might overlook many insights to the human visual system.

2. Since the gaze crop consists of almost half of the scene (224/336), and the gaze location is heavily biased to the center (Fig. 10 of the appendix), I am not sure if a gaze location is necessary to crop the frame. What would the results look like if the crop was a 224X224 or 336X336 box centered in the frame?

3. I am not convinced that a biologically plausible method would excel at fine-grained recognition but not as much at recognizing the coarser class labels. The analysis of the same is very hand-wavy in L.304-310, and overlooks the very small or non-existent gains in performance with the bio-inspired learning method.

**Questions:**

1. The authors talk about central vision, i.e., the part of the scene that is fixated, and in L.449, they mention that the crop is parafovea-sized can they elaborate how they map the gaze crop size in pixels to visual angle to infer that it is parafovea-sized?

2. What are the linear probe accuracies as reported in Table 1 for the ablation "w/o Central Vision" (provided in Table 2)?

---

> ### Author Response · Authors · 2025-11-21
>
> We thank the reviewer for the thoughtful and constructive feedback. Below we address the raised questions (Q1-2) and
> weaknesses (W1-3) and we indicate the corresponding changes in the revised manuscript.
>
> ---
>
> ### W1 — Rationale for Focusing on Central Vision
>
> We thank the reviewer for raising this point. Prior work has shown that central vision is the primary driver of semantic
> object representations in the visual cortex (as reviewed in the Introduction). Our goal in this paper is therefore to
> isolate the specific contribution of central vision to semantic object learning under temporal slowness, rather than to
> model the full complexity of processing central and peripheral visual input. We agree that such a more complete model
> could reveal additional interactions between high-acuity foveal input and lower-acuity peripheral input. This is an
> exciting direction for future work.
>
> ### W2 — Additional Analyses on Cropping Strategy
>
> We thank the reviewer for asking about the effect of using a centered crop rather than a gaze-based crop. Section 4.2
> includes experiments comparing our gaze-crop strategy to a crop centered in the frame with ViT-B/16 and ResNet50. Thanks
> to the reviewer’s comment, we now realize that these experiments were not sufficiently well exposed; we decided to move
> them to a new dedicated table, Table 2, in Section 4.2. We copy it here as well for convenience:
>
> | Semantic Group  |  ResNet50 CC  |  ResNet50 **GC**  |  ViT-B/16 CC  |  ViT-B/16 **GC**  |
> |:----------------|:-------------:|:-----------------:|:-------------:|:-----------------:|
> | Category        |     46.58     |     **46.94**     |     46.53     |     **46.76**     |
> | Fine-grained    |     37.77     |     **38.42**     |     50.31     |     **50.35**     |
> | Instance        |     62.83     |     **67.00**     |     67.99     |     **69.03**     |
>
> Across all semantic groups, the gaze-crop method (GC columns in the above table) consistently outperforms an equivalent
> center crop (denoted as CC in the table). This aligns with our ConvNeXt-B results in Appendix D1.
>
> ### W3 — Performance Differences between fine-grained and coarse category recognition
>
> We appreciate the reviewer’s concern. Our results clearly show that bio-inspired learning improves fine-grained object
> recognition. In Table 1, the average accuracy improves by 4,58% and 1,02% for ResNet50 and Vit-B/14, respectively.
> Additional results with ConvNeXt-B (Appendix D4, Table 4) confirm this trend.
>
> We acknowledge that the improvements reported in Table 1 for ImageNet-based datasets appear modest.
>
> First, for clarity and consistency, we reported results in Table 1 using the hyperparameter setting that performed best
> on average across all tasks. However, as discussed in Section 4.1, the optimal hyperparameters vary across semantic
> groups. This is particularly evident in Tables 8 and 9 (Appendix), where adjusting parameters such as the crop size (
> N=336) or the time window (ΔT=1) further improves performance on ImageNet datasets with bio-inspired learning (by
> +0.5–1%). Importantly, bio-inspired learning continues to yield significant gains for fine-grained and instance-level
> recognition, compared to our baseline.
>
> Second, in Section 4.2, we show that saccadic eye movements disrupt the temporal continuity of visual inputs,
> introducing noise into the temporal-slowness learning signal. This degradation ultimately harms the object-recognition
> performance of our bio-inspired model. It is, in our view, non-obvious that a biologically inspired learning rule would
> still support strong visual abilities in the presence of such disruptions. Moreover, when we suppress these
> saccade-induced interferences, we find that bio-inspired learning yields an additional improvement of up to +1.3% in
> category recognition, with gains of similar magnitude for fine-grained and instance-level recognition. These results
> suggest that humans’ knowledge of their own eye movements, together with the modeled learning mechanism, may
> synergistically strengthen emerging object representations.
>
> ### Q1 — Mapping crop size to parafoveal visual angle
>
> We thank the reviewer for pointing out this ambiguity. We dropped the statement that the crop is “parafovea-sized.”
> Because the dataset provides no camera intrinsics, we cannot estimate degrees of visual angle or map pixels to visual
> field size. We clarified this limitation in the revised manuscript. For completeness, we note that the crop is likely
> larger than a parafoveal window.
>
> (continued below)

---

> > ### Author Response · Authors · 2025-11-21
> >
> > ### Q2 — Linear probe accuracies for Control Condition
> >
> > The linear probe accuracies of our bio-inspired model and its ablation “w/o Central Vision” are as follows:
> >
> > | Semantic Group  |  ResNet50  Bio-inspired  |  ResNet50  w/o Central Vision  |  ViT-B/16  Bio-inspired  |  ViT-B/16  w/o Central Vision  |
> > |:----------------|:------------------------:|:------------------------------:|:------------------------:|:------------------------------:|
> > | Category        |        **46.94**         |             46.25              |        **46.76**         |             46.10              |
> > | Fine-grained    |        **38.34**         |             33.76              |        **50.35**         |             48.29              |
> > | Instance        |        **67.00**         |             64.69              |        **69.03**         |             66.11              |
> > | Scene           |          42.95           |           **44.27**            |          41.45           |           **43.43**            |
> >
> > The requested results were already in the manuscript. Figure 2 (gaze crop size = 540) reports the linear-probe
> > accuracies for the w/o Central Vision ablation. The corresponding numerical values are also provided in Table 8 (
> > Appendix). We now reference the former location explicitly in the revised version of Section 4.3.
> >
> > ---
> > We believe that the revisions and clarifications strengthen the manuscript, making its contributions clearer. We thank the reviewer for their thoughtful engagement with our work.

---

> > > ### Comment · Area_Chair_wore · 2025-11-22
> > >
> > > Hi Reviewer,
> > >
> > > The authors have submitted their responses to your reviews. Please take a look and let the authors know if you have any further questions or concerns. Thank you again for your contributions to ICLR!
> > >
> > > Best regards,
> > > AC

---

> > > > ### Comment · Reviewer_KB1f · 2025-11-22
> > > > **Thanks for the clarifications**
> > > >
> > > > Dear Authors,
> > > >
> > > > Thanks for the clarifications - they are sufficient for my review, and I have raised my score. Please incorporate the modifications suggested by all reviewers into your manuscript. Good luck!

---

### Author Response · Authors · 2025-11-21

Dear Reviewers and Area Chair,

We have uploaded revised versions of both the main paper and the supplementary material. All modifications made in response to the reviewers’ comments are now incorporated, and every change is **highlighted in blue** for clarity.

Best regards,

The Authors

---

### Meta-Review · Area_Chair_9U11 · 2025-12-29

**Summary:**

This paper investigates how humans might learn to recognize objects by combining two biological mechanisms: one is focusing on central vision (where the eyes look) and another is leveraging temporal slowness. The authors simulate this process using the existing Ego4D dataset, a gaze prediction model to mimic eye movements, and a self-supervised learning model trained on these gaze-centred crops. The results demonstrate that combining these mechanisms improves the model's ability to learn semantic object representations, particularly for fine-grained categorization and instance recognition, compared to baselines that use the full field of view or random crops. Although the method performs worse on scene recognition because it removes background context, the work offers a good scientific contribution by computationally validating biological learning theories. AC finds the motivation strong and the experimental analysis sufficient, supporting the claim that fixational eye movements and central vision are key drivers for learning object semantics. Therefore, I recommend accepting this work despite most of borderline scores.

**Reviewer Concerns:**

The authors did a good job during the rebuttal. They have answered the main questions raised during the discussion, particularly regarding whether the predicted eye movements were reliable. They satisfied Reviewers `KB1f`, `pvCu`, and `mtJa` by showing that models trained on predicted gaze performed similarly to those trained on real human eye-tracking data. To address doubts about whether the method was just simple cropping, the authors added new experiments proving their approach works better than fixed center crops or standard saliency models. While Reviewer `Qmz9` argued the method lacked technical complexity and did not compare enough with state-of-the-art engineering models, the authors clarified that their goal was to validate a biological theory rather than beat performance benchmarks.

**Reviewer Scores:**

Reviewer `KB1f` decided to raise the score. Reviewers `pvCu` and `mtJa` maintained their scores of 6 and confirmed their concerns were resolved once the authors provided the requested comparisons to other cropping methods. Reviewer `Qmz9` gave a low score of 2 and likely kept it, as they felt the work was too simple and did not offer enough technical novelty compared to existing methods.

---

### Decision · Program_Chairs · 2026-01-26

Accept (Poster)